# Cognitive experience alters cortical involvement in goal-directed navigation

Charlotte Arlt, Roberto Barroso-Luque, Shinichiro Kira, Carissa A Bruno, Ningjing Xia, Selmaan N Chettih, Sofia Soares, Noah L Pettit, Christopher D Harvey*

Department of Neurobiology, Harvard Medical School, Boston, United States

**Abstract** Neural activity in the mammalian cortex has been studied extensively during decision tasks, and recent work aims to identify under what conditions cortex is actually necessary for these tasks. We discovered that mice with distinct cognitive experiences, beyond sensory and motor learning, use different cortical areas and neural activity patterns to solve the same navigation decision task, revealing past learning as a critical determinant of whether cortex is necessary for goal-directed navigation. We used optogenetics and calcium imaging to study the necessity and neural activity of multiple cortical areas in mice with different training histories. Posterior parietal cortex and retrosplenial cortex were mostly dispensable for accurate performance of a simple navigation task. In contrast, these areas were essential for the same simple task when mice were previously trained on complex tasks with delay periods or association switches. Multiarea calcium imaging showed that, in mice with complex-task experience, single-neuron activity had higher selectivity and neuron–neuron correlations were weaker, leading to codes with higher task information. Therefore, past experience is a key factor in determining whether cortical areas have a causal role in goal-directed navigation.

## Editor's evaluation

The paper is of interest to people interested in understanding the neural substrates of learning and how these can be impacted by previous knowledge and training history. It also has relevance for behavioral neuroscientists when considering possible order effects of experiments.

*For correspondence: harvey@hms.harvard.edu

Competing interest: The authors declare that no competing interests exist.

## Introduction

The activity of neurons in the mammalian cortex and its correlation with aspects of sensation, cognition, and action have been studied extensively using decision tasks, but the factors that determine whether cortical areas are actually necessary for these tasks are not fully understood (*Gold and Shadlen, 2007*; *Hanks et al., 2006*; *Katz et al., 2016*; *Salzman et al., 1990*). Across studies, the necessity of cortical areas has been tested during tasks that involve different sensory, behavioral, and cognitive features (*Ceballo et al., 2019*; *Erlich et al., 2011*; *Fischer et al., 2020*; *Goard et al., 2016*; *Guo et al., 2014*; *Harvey et al., 2012*; *Inagaki et al., 2018*; *Licata et al., 2017*; *Raposo et al., 2014*; *Yang and Zador, 2012*; *Zhou and Freedman, 2019*; *Znamenskiy and Zador, 2013*). Collectively, these studies have formed proposals on the types of tasks for which specific cortical areas are essential. For example, in rodents, posterior parietal cortex (PPC) is necessary for visual but not some auditory discrimination tasks (*Licata et al., 2017*) and is considered to be especially involved in tasks that have a short-term memory component, such as a delay period between sensory cues and choice reports, or a requirement for evidence accumulation over time (*Lyamzin and Benucci, 2019*). As another example, area LIP in monkeys is thought to be essential for sensory but not motor aspects of

visual motion discrimination tasks (*Zhou and Freedman, 2019*). Notably, these studies have focused on how specific features of a task-of-interest determine which cortical areas are causally involved. However, in addition to the task-of-interest in a study, individual animals have learned a variety of associations throughout their lifetime and may have performed a diversity of tasks previously, often with different experiences between individuals. Although it is intuitive that past learning, beyond the demands of the task-of-interest, may impact how individuals perform decision tasks, most studies have not investigated the effect of past learning on the involvement of cortex. It therefore is not well understood how learning of previous tasks affects the necessity of cortical activity.

Two previous studies investigated this topic in the sensory domain, comparing the involvement of visual area MT in depth and motion perception in monkeys with different perceptual experience (*Chowdhury and DeAngelis, 2008*; *Liu and Pack, 2017*). For coarse depth discrimination, MT was only necessary in monkeys that had no prior experience in fine depth discrimination tasks (*Chowdhury and DeAngelis, 2008*), showing a decrease in cortical involvement with additional experience. In contrast, for motion discrimination, previous training on moving dot stimuli rendered MT necessary for discriminating the motion of gratings (*Liu and Pack, 2017*), showing that sensory experience can also increase cortical involvement. Relatedly, studies of motor learning have shown that cortex is essential during the learning process but becomes dispensable after learning is completed (*Hwang et al., 2019*; *Kawai et al., 2015*). In these cases, the animal's prior training was largely based on sensory or motor experience. However, studies have not investigated as extensively the impact of 'cognitive experience' on cortical necessity, which we broadly define as learning that extends beyond sensory or motor learning and includes learning of task rules and associations.

To explore the effects of cognitive experience on cortical necessity during decision tasks, we used the well-studied paradigm of spatial navigation, in which mice use sensory cues to navigate toward goal locations. Specifically, we used a virtual reality Y-maze in which mice encounter one of two visual cues and use the cue to virtually navigate to the left or right Y-arm to receive a reward (*Harvey et al., 2012*). This paradigm involves decision-making because it requires mice to choose one of two Y-arms based on a learned association with a visual cue. Furthermore, it involves navigation because mice must execute a sequence of movements to reach a goal location in a virtual environment, and movement through virtual mazes is known to recruit navigation-related circuits (*Aronov and Tank, 2014*; *Domnisoru et al., 2013*; *Harvey et al., 2009*). We therefore refer to our paradigm as goal-directed navigation, which implicitly includes decision-making to choose a goal location. We focused on two cortical association areas implicated in goal-directed navigation tasks, PPC and retrosplenial cortex (RSC) (*Driscoll et al., 2017*; *Fischer et al., 2020*; *Goard et al., 2016*; *Harvey et al., 2012*; *Morcos and Harvey, 2016*). PPC has been studied extensively in the context of decision-making across species and is generally thought to convert sensory cues into motor plans (*Freedman and Ibos, 2018*). Moreover, a comparison of several inactivation studies in rodents suggested that specific task components relying on short-term memory may determine whether PPC is necessary for a given decision task (*Lyamzin and Benucci, 2019*), offering a suitable entry point for exploring the effects of cognitive experience. PPC is strongly interconnected with RSC, which is considered to be critical for reference frame transformations underlying basic navigation and to support spatial memory across various timescales (*Alexander et al., 2020*; *Alexander and Nitz, 2015*; *Fischer et al., 2020*; *Milczarek et al., 2018*; *Pothuizen et al., 2009*; *Powell et al., 2017*). RSC is furthermore involved in cognitive processes beyond spatial navigation, such as context-dependent cue–choice associations, temporal learning, and value coding (*Franco and Goard, 2021*; *Hattori et al., 2019*; *Todd et al., 2015*), making it an especially interesting area to study in the context of cognitive experience.

To investigate the impact of cognitive experience on the necessity of these areas, we trained mice to perform different variants of a navigation task in virtual reality. One group of mice was trained only on a simple navigation task, whereas other groups were trained on complex tasks before being transitioned to the same simple task. The complex tasks involved the same visual cues and choice reports as the simple task but contained additional delay periods or association switches hypothesized to account for PPC involvement (*Lyamzin and Benucci, 2019*). This ensured that sensory and movement aspects of the complex and simple tasks were as identical as possible and allowed us to test the effect of 'cognitive experience' instead of perceptual or motor learning. Thus, this design tested the effect of cognitive experience on already learned associations and is potentially distinct from how cognitive experience may affect learning of new tasks involving novel associations, as in schema learning

(*Bartlett, 1932*; *McKenzie et al., 2013*; *Piaget, 1926*; *Tse et al., 2007*) and learning sets (*Caglayan et al., 2021*; *Eichenbaum et al., 1986*; *Harlow, 1949*).

We discovered that mice with different previous task experience used distinct sets of brain areas to solve the same simple task. During the simple navigation task, mice without prior training on complex tasks performed well above chance levels when RSC or PPC was inhibited using optogenetics. In contrast, during the same simple task, mice with prior complex-task training performed close to chance levels when these areas were inhibited. In addition, calcium imaging revealed that prior complex-task experience resulted in increased selectivity of neural activity patterns for task-relevant variables and decreased correlations in neural activity during the simple task. Thus, individuals with distinct cognitive experience make outwardly identical navigation decisions using different combinations of brain areas and neural activity patterns. We suggest that, because neural circuits are optimized for a wide range of computations beyond the ones required by a current task-of-interest, a global set of constraints and optimizations can dramatically impact the cortical areas that are necessary during goal-directed navigation.

## Results

### Increased necessity of cortical association areas in complex versus simple tasks

We developed a paradigm in which head-restrained mice running on a spherical treadmill were trained to use visual cues to make navigation decisions in a virtual reality Y-maze (*Figure 1A, B*). We used this paradigm to create a 'simple task' and two 'complex tasks'. In the simple task, mice learned to associate visual cues – horizontal and vertical bars – with left and right turns, respectively, to obtain rewards. In this task, the visual cue was present throughout the entire Y-maze, and the rewarded cue–choice associations (e.g., horizontal bars-left choice) did not change (*Figure 1C*).

The complex tasks were designed based on the same Y-maze concept and used the identical horizontal and vertical bars as visual cues. In the 'delay task', the visual cues were only present at the beginning of the maze, followed by a neutral visual pattern on the walls for the remainder of the maze (*Figure 1E*). This neutral pattern was identical across trials and did not provide information about the reward location. We chose this task design because it is common in decision tasks to insert a delay period between the sensory cues and choice reports and because this task has previously been shown to require activity in PPC (*Driscoll et al., 2017*; *Harvey et al., 2012*; *Pinto et al., 2019*). Mice trained on the delay task were initially trained on the simple task, and then the delay was introduced and gradually increased across sessions (*Figure 1—figure supplement 1*). Interestingly, mice apparently made decisions early in the trial even before the onset of the delay segment, as suggested by an early divergence in running trajectories for left and right choices (*Figure 1—figure supplement 2*). Nevertheless, task performance was lower in the delay task than the simple task, suggesting that the additional delay period increased task difficulty (*Figure 1K*).

We created another complex task termed the 'switching task', approximating more dynamic real-world environments in which associations between cues and rewarded actions change frequently. In the switching task, the rewarded relationships between the visual cues and left–right choices were switched across blocks several times within a single session, resulting in two rules (Rules A and B) (*Figure 1G*). The same visual cue was thus associated with left choices in one rule block and right choices in the other rule block. The value of each cue and action was constant across switches, in contrast to tasks with changing reward probabilities. The current rule and rule switch were not explicitly signaled, so mice inferred rule switches by trial and error and stored a belief of the current rule in short-term memory. Accordingly, after rule switches, performance dropped and then recovered to high and stable levels after tens of trials (*Figure 1G*), with similar performance recovery after both rule switches (*Figure 1—figure supplement 3*). For one of the two rules (Rule A), the trials were designed to be exactly the same as in the simple task, including the same cues and rewarded choices. In fact, the software code used to create the virtual environments was identical between the simple task and Rule A of the switching task. We chose this task design because rule switches are commonly used in decision tasks (*Johnston et al., 2007*; *Saez et al., 2015*; *Siniscalchi et al., 2016*), and the switch allowed us to preserve identical trials between Rule A and the simple task.

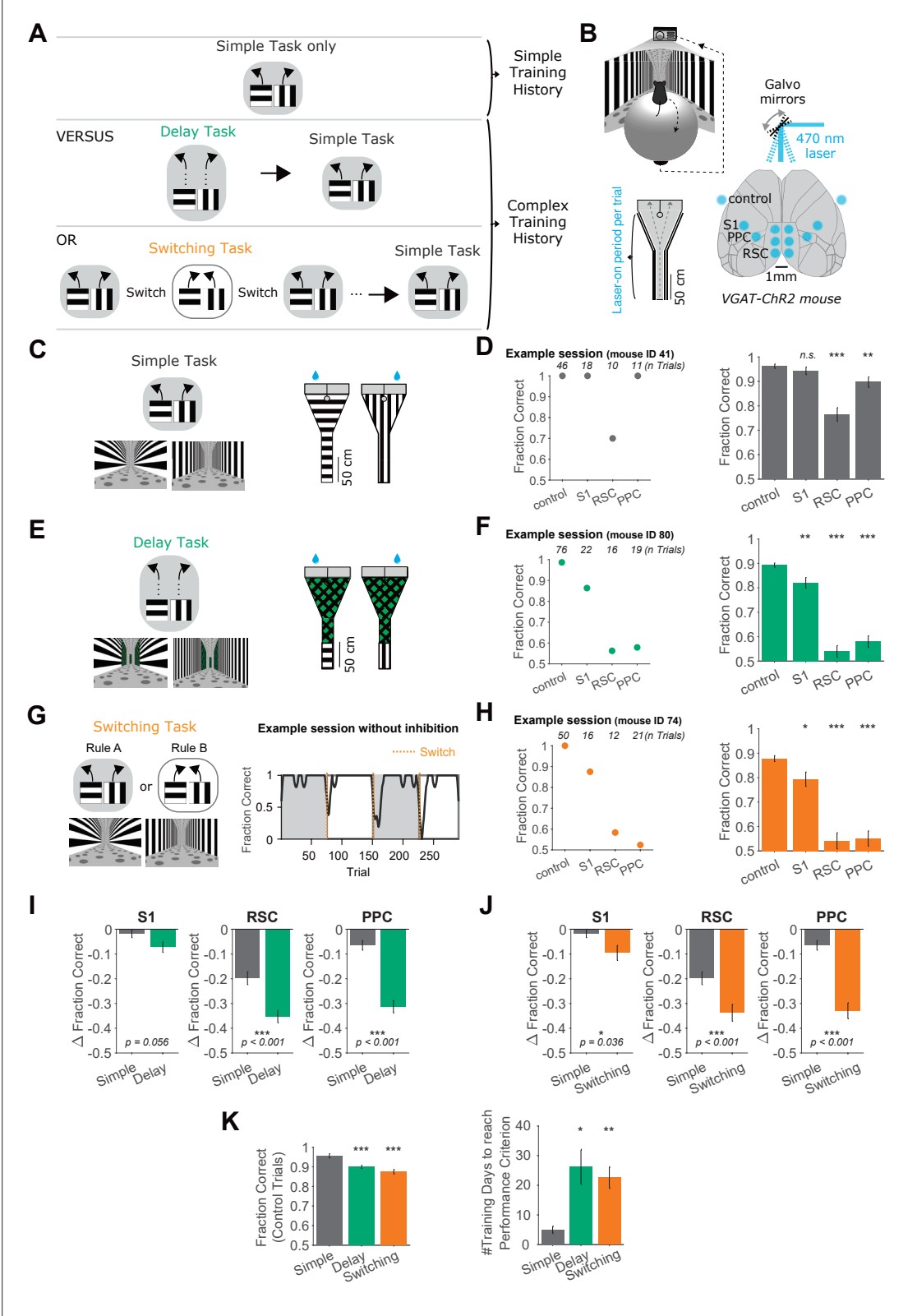

**Figure 1.** Increased necessity of cortical association areas in complex versus simple decision tasks. (**A**) Schematic overview of the behavioral tasks and task training sequences used in this study. Top row: One group of mice is trained in the simple task only. Middle row: Another group of mice is trained on the delay task and then transitioned to the simple task. Bottom row: Another group of mice is trained on the switching task and then transitioned to the simple task. The middle and bottom rows indicate complex training histories. (**B**) Top: Schematic of virtual reality behavioral setup. Bottom right:

*Figure 1 continued*

Schematic of optogenetic inhibition with bilateral target locations. Bottom left: Top view of Y-maze. Inhibition lasted from trial onset throughout maze traversal. (**C**) Left: Simple task schematic indicating two trial types (horizontal or vertical cues) and corresponding rewarded navigation decisions (running left or right). Corresponding VR screenshots at the trial start are below. Right: Top view of the two maze schematics. Water drops indicate hidden reward locations. (**D**) Left: Example session in the simple task showing mean performance for each inhibited location. Right: Performance in the simple task for each inhibited location across 45 sessions from 4 mice. Bars indicate mean ± standard error of the mean (SEM) of a bootstrap distribution of the mean. S1 p = 0.84; RSC p < 0.001; PPC p = 0.006; from bootstrapped distributions of ΔFraction Correct (difference from control performance) compared to 0, two-tailed test, α = 0.05 plus Bonferroni correction. *: p < 0.05; **: p < 0.01; ***: p < 0.001. Sessions per mouse: 11 ± 2. Trials per session: 53 ± 23 (control), 19 ± 8 (S1), 18 ± 9 (RSC), 20 ± 9 (PPC), mean ± standard deviation (SD). (**E**) Similar to (**C**), but for the delay task. (**F**) Similar to (**D**), but for the delay task. Sixty-two sessions from 7 mice. S1 p = 0.006; RSC p < 0.001; PPC p < 0.001. Sessions per mouse: 9 ± 4. Trials per session: 60 ± 15 (control), 16 ± 6 (S1), 15 ± 4 (RSC), 17 ± 5 (PPC), mean ± SD. (**G**) Left: Schematic of the switching task, utilizing the identical mazes as the simple task. The cue–choice associations from the simple task (Rule A) were switched within a session (to Rule B). Right: Behavioral performance from an example session. Dotted orange lines indicate rule switches. (**H**) Similar to (**D**), but for the switching task, Rule A trials only. 89 sessions from 6 mice. S1 p = 0.036; RSC p < 0.001; PPC p < 0.001. Sessions per mouse: 15 ± 5. Trials per session: 26 ± 9 (control), 8 ± 3 (S1), 7 ± 4 (RSC), 8 ± 3 (PPC), mean ± SD. (**I**) Comparison of inhibition effects (ΔFraction Correct) in the simple and the delay tasks for each cortical inhibition location. Bars indicate mean ± SEM of a bootstrap distribution of the mean; two-tailed comparisons of bootstrapped ΔFraction Correct distributions, α = 0.05. *: p < 0.05; **: p < 0.01; ***: p < 0.001. Same datasets as in (**F**, **G**). (**J**) Similar to (**I**), but for the simple versus switching task (Rule A trials only). Same datasets as in (**F**, **H**). The simple task data are the same as in (**I**). (**K**) Left: Comparison of performance on control trials across tasks, using only the first two laser-on blocks in each session. Bars indicate mean ± SEM of a bootstrap distribution of the mean. Delay versus simple p < 0.001; switching versus simple p < 0.001; two-tailed comparisons of bootstrapped Fraction Correct distributions, α = 0.05. *** p < 0.001. Right: The number of training sessions needed to reach performance criteria across tasks (Methods). Bars indicate mean ± SEM across mice, n = 4 for simple task, n = 5 for delay task, n = 6 for switching task. Both delay and switching task data were compared to the simple task data using an unpaired two-sided *t*-test. Delay versus simple p = 0.015; switching versus simple p = 0.006. *: p < 0.05, **: p < 0.01.

The online version of this article includes the following figure supplement(s) for figure 1:

**Figure supplement 1.** Behavioral training stages and task transitions.

**Figure supplement 2.** Choice decoding from running, stability of inhibition effects and choice biases from inhibition across tasks.

**Figure supplement 3.** Performance after rule switches and inhibition effects in high-performance periods of Rule B in the switching task.

**Figure supplement 4.** Similar deficits from inhibition in a run-to-target task as in the simple task.

**Figure supplement 5.** Cue only or delay only inhibition in the delay task.

**Figure supplement 6.** Performance on trials following inhibition and rule switching with inhibition during the feedback/ITI period.

Across the simple and two complex tasks, mice experienced the same choice-informative sensory cues and had to run through similar or identical mazes to report their choices. Thus, the key difference between the simple and complex tasks was due to 'cognitive' complexity through the addition of a delay period or frequent switches in the associations, rather than differences in the sensory cues informing choices or differences in motor output.

We tested the necessity of various cortical areas during the simple and complex tasks using optogenetics to activate GABAergic interneurons, which leads to silencing of nearby excitatory neurons (***Guo et al., 2014***; ***Li et al., 2019***; ***Minderer et al., 2019***). Channelrhodopsin-2 (ChR2) was expressed in inhibitory interneurons in transgenic mice and photostimulated using a clear-skull preparation with a two-dimensional laser scanning system (***Figure 1B***). We focused inhibition on two cortical association areas previously linked to decision-making and navigation, PPC and RSC (***Driscoll et al., 2017***; ***Fischer et al., 2020***; ***Harvey et al., 2012***). To match each area's anatomical extent, we used three inhibition spots in each hemisphere for RSC and one spot per hemisphere for PPC. As a control, we inhibited a spot in primary somatosensory cortex (S1), an area not implicated in visual decision-making (***Figure 1B***). Inhibition trials were interleaved with control trials in which the laser spot was positioned outside the mouse's brain. On inhibition trials, photostimulation was applied bilaterally and throughout the duration of the mouse's maze traversal.

We first considered inhibition effects on performance of the simple task in mice that had only been trained in the simple task (***Figure 1D***). Silencing S1 did not affect performance, and inhibition of PPC resulted in a very small performance decrease, with mean performance of 90 ± 2% correct (mean ± standard error of the mean [SEM]). Inhibition of RSC had the largest effect, resulting in intermediate performance levels of 77 ± 3% correct. However, mice still performed well above chance (50% correct). To assess whether the effect of RSC inhibition was specific to the cognitive requirements of the simple task or related to lower-level processes required for task performance such as vision,

movement, and basic navigation, we silenced the same areas in an even simpler task in which mice ran toward a visual target present on either side of the maze end to obtain rewards (*Harvey et al., 2012*). Effects on performance were similar in this run-to-target task, suggesting that RSC inhibition in the simple task may impair lower-level processes such as basic navigation instead of decision-making based on cue–choice associations (*Figure 1—figure supplement 4*). Together, these results indicate that the cortical areas we silenced were only modestly involved in the simple task.

We next considered the effects of inhibition on mice performing the delay task or the switching task (*Figure 1E-H*). In the switching task, we silenced cortical areas during the periods of high performance after accuracy had recovered following a rule switch (*Figure 1G*) and initially focused analysis only on Rule A trials, as they are the exact same trials as present in the simple task. S1 inhibition during the complex tasks caused a modest decrease in performance, but mice still performed the tasks at high levels (*Figure 1F, H*). In contrast, inhibition of PPC or RSC greatly impaired performance in the delay and switching tasks and resulted in performance of 55 ± 3% correct, which is close to chance levels. With PPC and RSC inhibition, many mice exhibited biases in the choices they made, whereas others appeared to choose randomly between left and right (*Figure 1—figure supplement 2*). The effects of inhibiting PPC and RSC were markedly larger in the delay and switching tasks than in the simple task (*Figure 1I, J*). Therefore, adding a delay epoch to the trial or frequent association switches across trials increased the necessity of cortex relative to the simple task. Similar results were observed when comparing the simple task to Rule B trials of the switching task (*Figure 1—figure supplement 3*). Furthermore, inhibition effects appeared stable across dozens of experienced sessions and were independent of how early in the maze a mouse's choice could be inferred from its running pattern (*Figure 1—figure supplement 2*).

We also asked whether the increased cortical necessity was especially apparent in the task period that was changed compared to the simple task. In the delay task, we thus restricted photoinhibition to the cue or the delay segment of the trial (*Figure 1—figure supplement 5*). Inhibition of PPC or RSC in both trial segments decreased performance to a greater degree than in the simple task, indicating that PPC and RSC are necessary for multiple epochs of the task. In the switching task, the main difference relative to the simple task is the introduction of association switches, so activity in PPC and RSC may be necessary for storing or updating the rule. Taking advantage of interleaved control and inhibition trials, we found that the effects of inhibition were restricted to the current trial, as inhibition did not affect subsequent control trials (*Figure 1—figure supplement 6*). We also looked for a role in updating the rule, which is likely critical following the completion of a trial, in particular after a rule switch. We focused on PPC as a candidate for updating the rule because PPC has sensory- and choice-related history signals (*Akrami et al., 2017*; *Morcos and Harvey, 2016*). We silenced PPC during the intertrial interval on every trial following a rule switch, for 50 consecutive trials. However, PPC inhibition did not affect the recovery of performance after a rule switch (*Figure 1—figure supplement 6*). Together, these results suggest that the large inhibition effects during the switching task are not due to impaired storing or updating of the rule. Thus, the increased necessity of PPC and RSC during the delay and switching tasks did not appear to be specific to the added task components.

Overall, our results indicate that more complex tasks require cortical activity, specifically in PPC and RSC, to a larger extent than a simple task. We verified that the complex tasks were indeed more challenging for mice. Relative to the simple task, it took mice longer to become experts at the delay and switching tasks (Methods), and their performance on control trials was lower (*Figure 1K*). These findings are consistent with, and extend, previous work that concluded cortical necessity increases with task complexity (*Harvey et al., 2012*; *Pinto et al., 2019*).

## The necessity of PPC and RSC depends on a mouse's previous cognitive experience

This set of tasks provided a platform for testing the effects of cognitive experience on cortical necessity for simple task performance, by comparing groups of mice with or without previous training on the complex tasks. A first group of mice was only trained on the simple task. The second group was first trained on one of the complex tasks and then transitioned to the simple task for 14 consecutive sessions (one session per day), without experiencing the complex task again. Different cohorts of mice were transitioned to the simple task from the switching and delay tasks (also see *Figure 1—figure supplement 1*). Note that both for mice trained on the delay and switching tasks, the simple task is

not a novel task in that it contains the same cue–choice associations as those learned in each complex task. This design allowed us to compare different mice performing the same task (the simple task) but with distinct training histories.

We first considered the mice that were trained to be experts on the delay task and then transitioned to the simple task (*Figure 2A*). In these mice, inhibition of PPC and RSC during the simple task greatly impaired behavioral performance to 67 ± 3% and 62 ± 2% correct (mean ± SEM), respectively (*Figure 2B, C*). Thus, these mice needed PPC and RSC activity to perform at high levels on the simple task. These results were strikingly different from the inhibition effects in mice trained only on the simple task. Without complex-task experience, mice performed at 90 ± 2% and 77 ± 3% correct with PPC or RSC inhibited (*Figure 2C*), respectively, indicating they did not strongly rely on PPC or RSC activity for task performance. In contrast, inhibition of S1 had little effect on performance in the simple task both with and without previous delay task training. The larger effect of PPC and RSC inhibition due to previous delay task experience was not only apparent immediately after the transition to the simple task but persisted for the full 2 weeks that we investigated (*Figure 2D, E*). Therefore, the effect of PPC and RSC inhibition had markedly larger effects in the simple task when mice had previous training in the delay task, both in the first and second weeks after the task transition. Thus, mice with distinct previous task experience require different cortical areas to perform the same task.

This persistent effect of previous task experience is particularly surprising because the task transition should be immediately visually apparent to mice due to the lack of a delay period in each trial. We wondered whether mice with delay task experience may find the simple task to be challenging or need to undergo a significant learning phase to perform the simple task at expert levels. However, these mice performed at very high levels on trials without laser light even in the first session after the transition to the simple task (*Figure 2—figure supplement 1*). Relatedly, performance on control trials in the simple task was as high in mice with delay task experience as in mice with simple task experience only, indicating that subjective task difficulty did not differ depending on training history (*Figure 2F*). These results support the notion that the simple task was not a newly learned task but rather the same associations as learned for the delay task.

We reached a similar conclusion when we compared mice performing the simple task with and without previous training on the switching task (*Figure 3A*). In mice that had previously been experts on the switching task, PPC and RSC inhibition resulted in performance of 67 ± 3% and 61 ± 3% correct (mean ± SEM), respectively, on the simple task (*Figure 3B, C*). As for previous training on the delay task, this effect of PPC and RSC inhibition during the simple task was markedly larger than when these areas were inhibited in mice without previous training on the switching task (*Figure 3D, E*). PPC activity was necessary even 2 weeks after the transition to the simple task. RSC's involvement was greatest in the first week after the transition. Therefore, mice with previous experience in the switching task require PPC and RSC activity to perform the simple task, whereas these areas are largely dispensable during performance of the same simple task in mice without this previous training.

We wondered if the mice transitioned from the switching task to the simple task might continue to behave as if they were in the dynamic context of the switching task, and thus may find the simple task more challenging than mice without switching task experience. However, it appeared that mice adapted behaviorally to the simple task quickly after the transition. As for mice with previous delay task experience, performance on trials without laser light was high even in the first session after the switching-to-simple task transition (*Figure 2—figure supplement 1*), and performance on control trials reached levels observed in mice without previous training on the switching task (*Figure 3F*). Thus, mice did not appear to require de novo learning of the simple task, as expected since the associations in the simple task are identical to those contained in the switching task. Also, in the switching task, performance at the start of sessions was only at intermediate levels as mice determined the current rule (*Figure 2—figure supplement 1*). In contrast, after a few days in the simple task, performance was near perfect even in the first tens of trials within a session. Interestingly, when presented with the opposite rule from the switching task again after 2 weeks on the simple task, mice could still switch back to the long unseen rule within a single session (*Figure 2—figure supplement 1*). Thus, although mice appeared to retain an understanding of potential association switches, their behavior did not reflect such expectations soon after they were transitioned to the simple task.

We also assessed whether the persistent increase of cortical necessity due to complex-task experience extended to the run-to-target task. Notably, inhibition of PPC and RSC during the run-to-target

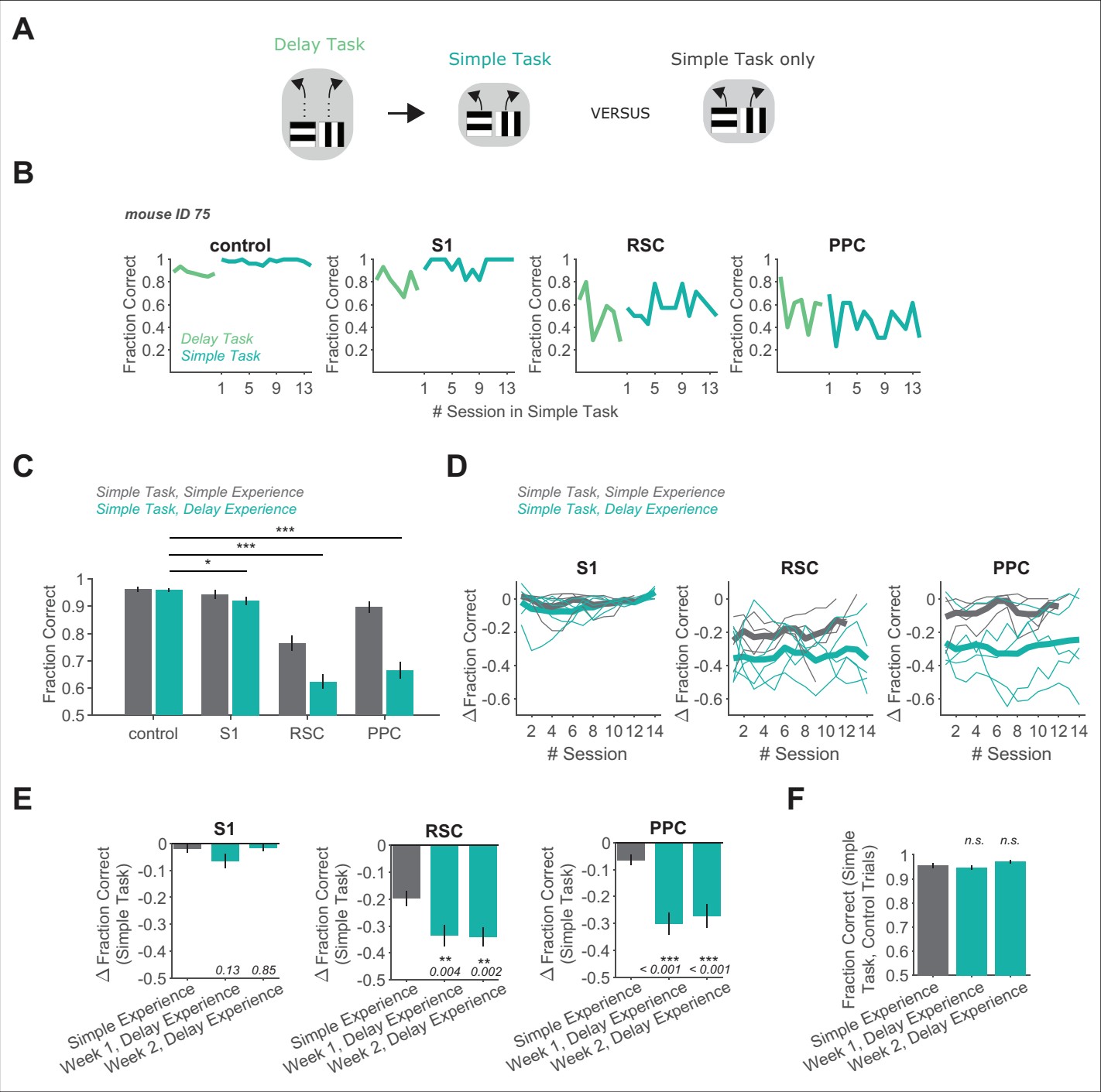

**Figure 2.** Delay task experience increases the necessity of RSC and PPC in a simple decision task. (**A**) Schematic of the training history sequence. One group of mice was trained on the delay task and then permanently transitioned to the simple task. This group of mice was compared to another group trained only on the simple task. (**B**) Performance of an example mouse transitioned from the delay task to the simple task on control and inhibition trials. (**C**) Performance in the simple task for each inhibited location in mice with simple task experience only (gray, 45 sessions from 4 mice, same dataset as in *Figure 1F*), and in mice with previous delay task experience (blue, 70 sessions from 5 mice). Bars indicate mean ± standard error of the mean (SEM) of a bootstrap distribution of the mean. S1 p = 0.012; RSC p < 0.001; PPC p < 0.001; from bootstrapped distributions of ΔFraction Correct (difference from control performance) compared to 0, two-tailed test, α = 0.05 plus Bonferroni correction. *: p < 0.05; ***: p < 0.001. Sessions per mouse: 14. Trials per session: 53 ± 7 (control), 13 ± 3 (S1), 13 ± 2 (RSC), 14 ± 2 (PPC), mean ± standard deviation (SD). (**D**) Inhibition effects (ΔFraction Correct) across sessions in the simple task in mice with only simple task experience (gray), and in mice with previous delay task experience (blue), for each cortical inhibition location. Thin lines show individual mice (n = 4 with simple task experience, n = 5 with delay task experience), thick lines show average across mice. ΔFraction Correct was smoothed with a moving average filter of three sessions. (**E**) Comparison of inhibition effects (ΔFraction Correct) in the

*Figure 2 continued on next page*

*Figure 2 continued*

simple task for mice with simple task experience only (45 sessions from 4 mice) versus delay task experience 1 or 2 weeks after transition from the delay task to the simple task (35 sessions per week from 5 mice). Bars indicate mean ± SEM of a bootstrap distribution of the mean; two-tailed comparisons of bootstrapped ΔFraction Correct distributions, α = 0.05. **: p < 0.01; ***: p < 0.001. Simple experience datasets are the same as in *Figure 1F* and *Figure 2C*. (F) Comparison of performance on control trials in the simple task with simple versus delay task experience, using only the first two laser-on blocks in each session. Bars indicate mean ± SEM of a bootstrap distribution of the mean. Simple task data in week 1 (p = 0.59) and week 2 (p = 0.19) after transition from the delay task were compared to the simple task only experience data; two-tailed comparisons of bootstrapped Fraction Correct distributions, α = 0.05. Trials per session: 51 ± 23 (simple experience), 51 ± 6 (delay experience, week 1), 53 ± 3 (delay experience, week 2), mean ± SD.

The online version of this article includes the following figure supplement(s) for figure 2:

**Figure supplement 1.** Mice with complex-task experience perform at high levels in the simple task.

task resulted in similarly minor performance drops in groups of mice with and without prior complex-task experience (*Figure 3—figure supplement 1*). Thus, previous experience in complex tasks did not make cortex essential for all simple tasks.

Collectively, these results highlight that the cortical areas used to perform a task can be profoundly shaped by experience from weeks ago. Mice with different previous task experience use distinct sets of cortical areas to solve the same task. Therefore, an understanding of which areas of cortex are necessary for decision tasks requires considering both the demands of the task-of-interest and the previous experiences of the individual.

## PPC and RSC neurons have activity patterns with higher selectivity in the switching task

Given that the necessity of cortical areas was modulated by previous training, we next asked if the neural activity patterns in these areas are also affected. Changes in an area's necessity from previous experience may or may not be accompanied by changes in neural activity. For instance, previous learning in the sensory domain changes cortical necessity, but not neural activity patterns, which can be explained by changes solely to how an area's activity is read out by downstream areas (*Chowdhury and DeAngelis, 2008*; *Liu and Pack, 2017*). Alternatively, neural activity patterns may also change along with an area's necessity, as observed in comparisons across different types of navigation tasks (*Harvey et al., 2012*; *Pinto et al., 2019*).

We simultaneously measured the activity of neurons in PPC, RSC, and primary visual cortex (V1) with two-photon calcium imaging using a large field-of-view, random-access microscope (*Sofroniew et al., 2016*; *Figure 4A–C*). This microscope allowed us to simultaneously image hundreds of neurons in each of these three cortical areas, with single-cell resolution. We focused our imaging on PPC and RSC because these areas showed major differences in necessity depending on previous experience. We also included V1 because it is densely interconnected with PPC and RSC (*Zhang et al., 2016*), contains cognitive, nonvisual signals in navigation decision tasks (*Koay et al., 2020*; *Saleem et al., 2018*), and is likely a key contributor during visual navigation. Note that we did not choose V1 as a target in our photoinhibition experiments because inhibition of V1 would likely cause visual processing deficits in all visual navigation tasks regardless of cognitive experience. We restricted our imaging experiments to the simple task and the switching task, given that they contain the identical virtual environments.

We first imaged neural activity in separate sets of mice in the simple and switching tasks (*Figure 4D*). Again, mice took longer to learn the switching task than the simple task (*Figure 4—figure supplement 1*). For a direct comparison of the switching and simple tasks, as was done for the inhibition experiments, we restricted analysis in the switching task to periods of high performance after performance had recovered following rule switches. While the trials around the rule switch are of interest generally, they may be challenging to compare to the simple task because performance is at different levels and mice might use distinct behavioral strategies after a rule switch versus during periods of high performance. To compare identical trial types across tasks, that is trials with the same cue–choice associations, we initially only considered Rule A trials in the switching task. We started by looking at a basic measure of neural activity, the overall level of activity in individual neurons. Interestingly, this basic measure revealed differences across tasks, as neurons in RSC and PPC had higher activity in the switching task than in the simple task (*Figure 4D*).

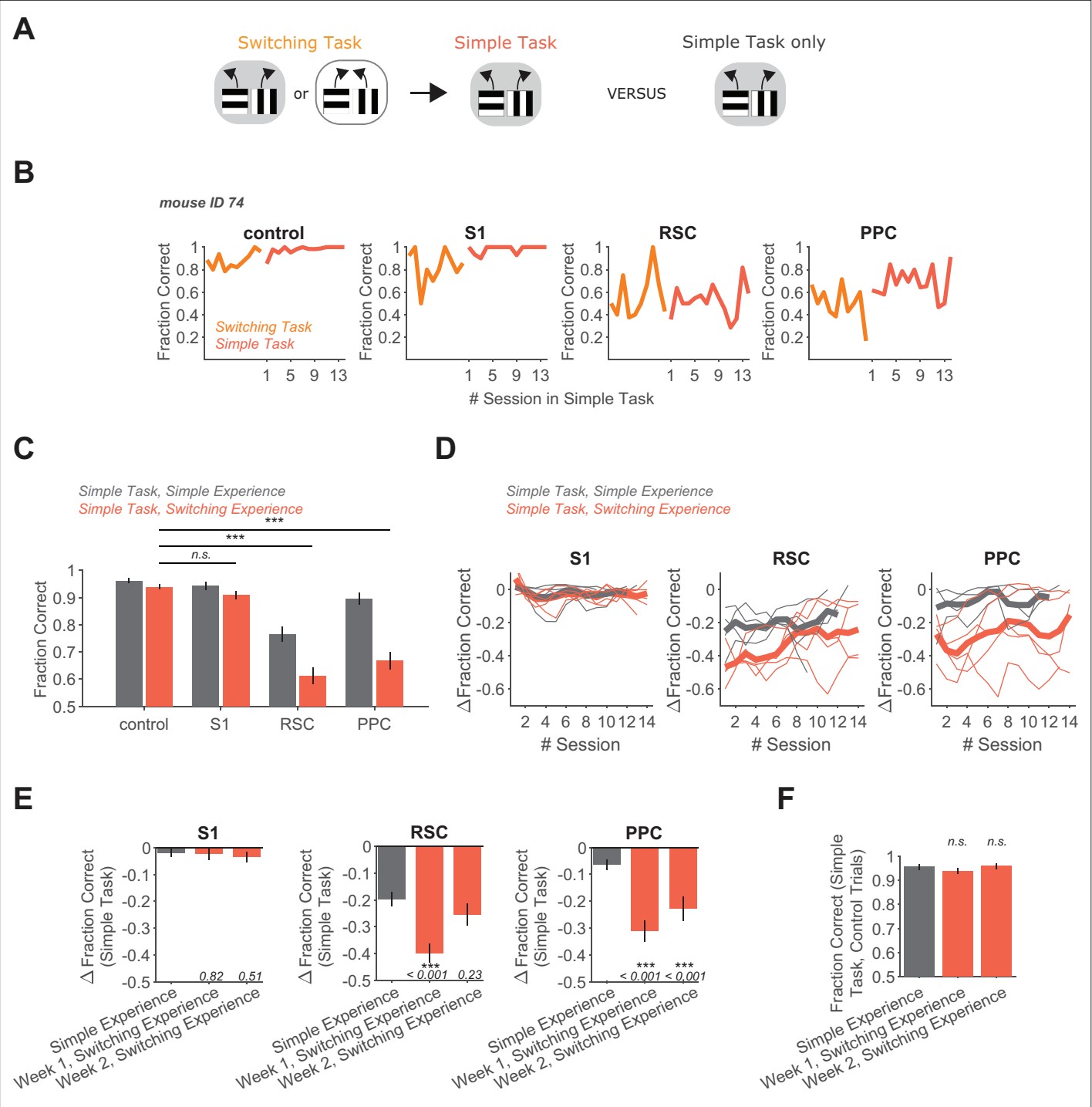

**Figure 3.** Switching task experience increases the necessity of RSC and PPC in a simple decision task. (**A**) Schematic of the training history sequence. One group of mice was trained on the switching task and then permanently transitioned to the simple task. This group of mice was compared to another group trained only on the simple task. (**B**) Performance of an example mouse transitioned from the switching task to the simple task on control and inhibition trials. (**C**) Performance in the simple task for each inhibited location in mice with simple task experience only (gray, 45 sessions from 4 mice, same dataset as in *Figure 1F*), and in mice with previous switching task experience (red, 69 sessions from 5 mice). Bars indicate mean ± standard error of the mean (SEM) of a bootstrap distribution of the mean. S1 p = 0.26; RSC p<0.001; PPC p < 0.001; from bootstrapped distributions of ΔFraction Correct (difference from control performance) compared to 0, two-tailed test, α = 0.05 plus Bonferroni correction. ***: p < 0.001. Sessions per mouse: 13 ± 0.4. Trials per session: 55 ± 11 (control), 14 ± 4 (S1), 13 ± 4 (RSC), 15 ± 4 (PPC), mean ± standard deviation (SD). (**D**) Inhibition effects (ΔFraction Correct) across sessions in the simple task in mice with only simple task experience (gray), and in mice with previous switching task experience (red), for each cortical inhibition location. Thin lines show individual mice (n = 4 with simple task experience, n = 5 with switching task experience), thick

*Figure 3 continued*

lines show average across mice. ΔFraction Correct was smoothed with a moving average filter of three sessions. (**E**) Comparison of inhibition effects (ΔFraction Correct) in the simple task for mice with simple task experience only (45 sessions from 4 mice) versus switching task experience 1 (35 sessions from 5 mice) or 2 (34 sessions from 5 mice) weeks after transition from the switching task to the simple task. Bars indicate mean ± SEM of a bootstrap distribution of the mean; two-tailed comparisons of bootstrapped ΔFraction Correct distributions, α = 0.05. ***: p < 0.001. Same datasets as in *Figure 1F* and *Figure 3C*. (**F**) Comparison of performance on control trials in the simple task with simple versus switching task experience using only the first two laser-on blocks in each session. Bars indicate mean ± SEM of a bootstrap distribution of the mean. Simple task data in week 1 (p = 0.32) and week 2 (p = 0.81) after transition from the switching task were compared to the simple task only experience data; two-tailed comparisons of bootstrapped Fraction Correct distributions, α = 0.05. Trials per session: 51 ± 23 (simple experience), 51 ± 5 (switching experience, week 1), 50 ± 7 (switching experience, week 2), mean ± SD.

The online version of this article includes the following figure supplement(s) for figure 3:

**Figure supplement 1.** Increased cortical involvement in the simple task after complex-task experience does not generalize to the run-to-target task.

Next, we considered that a direct way a cortical area may contribute to the task is by having activity that is different for the two trial types containing distinct cue–choice associations. Trial-type selectivity is a common measure for neural correlates of decision-related functions because it would allow a downstream area to read out the identity of the association and to execute the appropriate choice. We measured this selectivity as our ability to identify the trial type based on a neuron's activity and quantified it as the area under the receiver operating characteristics curve (auROC, *Figure 4E*). PPC and RSC neurons showed higher average levels of trial-type selectivity in the switching task than in the simple task (*Figure 4F*). At the level of populations of neurons, the trial-type selectivity was structured as sequences of neural activity, in which individual neurons were transiently active and different neurons were active at different locations along the maze (*Figure 4—figure supplement 1*). In addition, the fraction of RSC or PPC neurons with significant trial-type selectivity was higher in the switching task than in the simple task (*Figure 4G*).

We next assessed how well the current trial type could be decoded from the activity of neural populations of varying sizes in each area. In the simple task, RSC and PPC populations contained task-relevant information that led to above-chance decoding from a population of neurons. However, for the same size population in the switching task, this decoding accuracy in RSC and PPC was even higher, in line with the observed increased selectivity and larger fraction of selective neurons relative to the simple task (*Figure 4H*). These differences in activity across tasks in RSC and PPC were especially striking because, in the simple task and Rule A of the switching task, mice ran through a maze with identical visual cues and made similar left–right behavioral choices in both tasks. Therefore, the activity levels and selectivity of single neurons are higher in PPC and RSC when mice perform a more complex task, even when the sensory stimuli and choice reports in the tasks are identical, leading to better ability to decode the trial type.

In contrast, V1 neurons had similar levels of activity, selectivity, and population-level trial-type decoding in the simple and switching tasks (*Figure 4D–H*). This finding is consistent with the identical visual scene in these tasks but is perhaps surprising given that V1 neurons have been shown to contain nonvisual signals (*Koay et al., 2020*; *Saleem et al., 2018*).

We verified that the differences in selectivity across tasks were not due to differences in running patterns. When we selected sessions so that the time course and magnitude of decoding the mouse's reported choice from its running were similar across tasks, we largely observed the same differences in neural trial-type selectivity as reported above (*Figure 4—figure supplement 2*). Thus, the differences in neural selectivity cannot be trivially explained by differences in running patterns.

We also compared activity in the simple task to Rule B trials in the switching task and found similar differences in mean activity and selectivity levels across tasks (*Figure 4—figure supplement 3*). Given that selectivity was higher in both rules of the switching task, we wondered how selectivity was organized across rules in the switching task. By examining single-neuron activity across all four trial types, that is cue–choice combinations, we found that rather than being generally cue-, choice-, or rule-selective, a large fraction of neurons in all imaged areas, but highest in RSC, showed increased activity specifically for a single trial type (*Figure 4—figure supplement 4*). Thus, neurons were selective for a specific cue–choice combination, which is consistent with a phenomenon often called nonlinear mixed selectivity (*Fusi et al., 2016*; *Rigotti et al., 2013*).

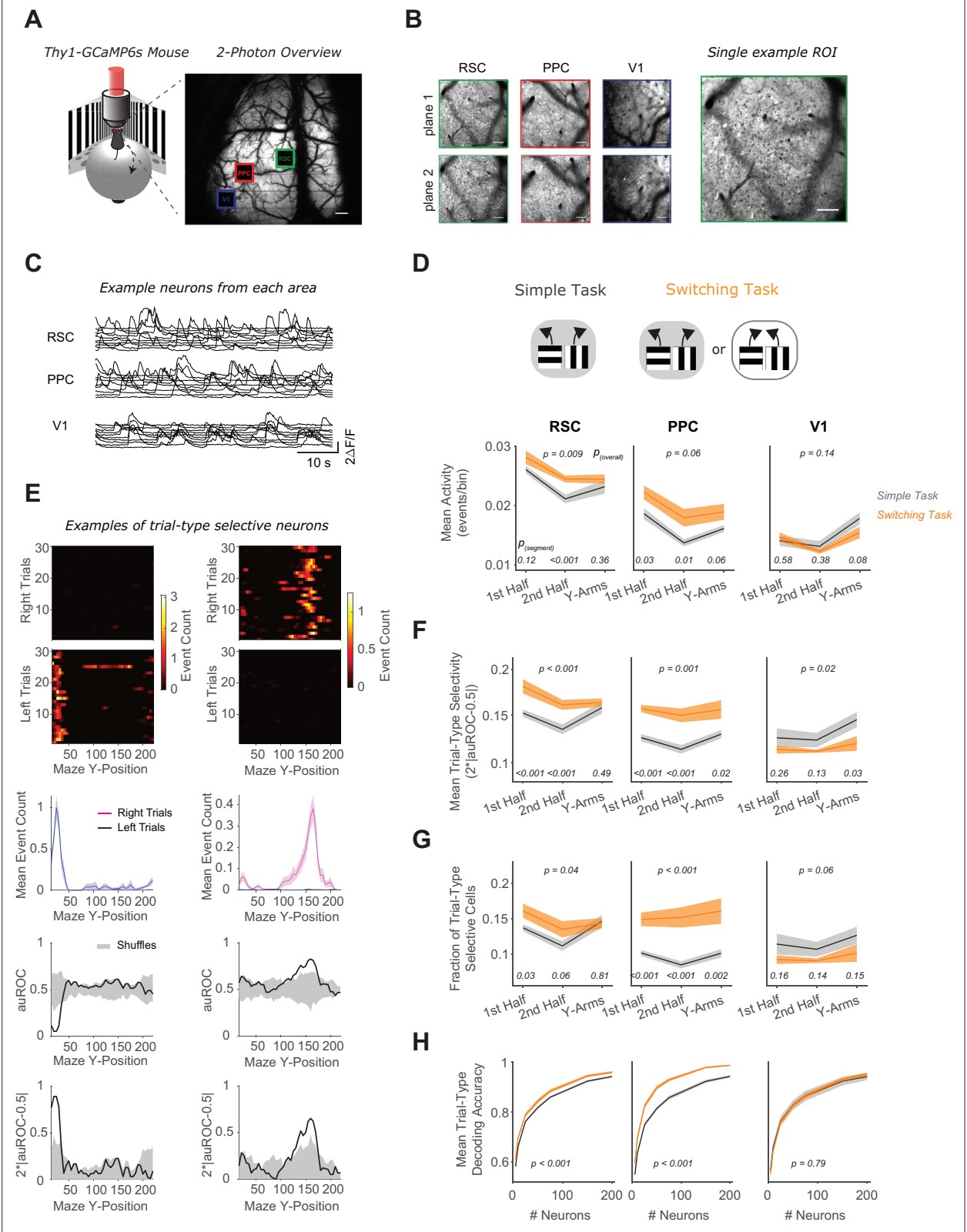

**Figure 4.** RSC and PPC neurons have activity patterns with higher selectivity in the switching task. (**A**) Left: Schematic of virtual reality behavioral setup with mesoscopic two-photon imaging. Right: Two-photon overview image of cortical window and locations of three areas imaged simultaneously. Scale bar: 500 μm. (**B**) Mean intensity two-photon images of each imaged area, color coded by area as in (**A**). Areas were imaged at two depths. Scale bars: 100 μm. (**C**) Example activity traces of 10 cells from each area. (**D**) In each area, mean activity levels across cells by maze segment are compared in the

*Figure 4 continued on next page*

*Figure 4 continued*

simple (gray) versus the switching (orange) task (only Rule A trials). The maze segments are the first half of the Y-stem (15–75 cm Y-Position), the second half of the Y-stem (75–150 cm), and the Y-arms (150–220 cm). Shading indicates mean ± standard error of the mean (SEM) of bootstrapped distributions of the mean. $p_{(segment)}$ shows p values of two-tailed comparisons of bootstrapped distributions per maze segment. $p_{(overall)}$ shows the p value for the task factor from a two-way analysis of variance (ANOVA; factors: task and maze segment). Simple task: $n = 3$ mice, 4 sessions per mouse, neurons per session by area: RSC: 1438 ± 217, PPC: 456 ± 172, V1: 498 ± 170. Switching task: $n = 3$ mice, 4 sessions per mouse, neurons per session by area: RSC: 1510 ± 398, PPC: 513 ± 304, V1: 363 ± 92 (mean ± standard deviation [SD]). (**E**) Left and right panel columns show activity from two different example neurons. Top: Spatially binned activity separated by trial type. Each row shows a single trial. Trials were subsampled to 30 trials per trial type. Middle: Mean activity for each trial type (pink: right trials, blue: left trials). Bottom: The area under the ROC curve (auROC) was calculated for each spatial bin. A shuffled distribution of auROC values (gray) was generated by randomly assigning left/right trial labels to each trial and recomputing auROC 100 times. Trial-type selectivity was defined as an absolute deviation of auROC from chance level (2*|auROC − 0.5|). To determine significance of trial-type selectivity, at each bin, this value was compared to the trial-type selectivity of the shuffle distribution (gray, significance threshold of $p < 0.01$). (**F**) Similar to (**D**), except for the metric of trial-type selectivity, that is 2*|auROC − 0.5|. (**G**) Similar to (**D**), except for the fraction of trial-type selective cells as determined from comparing each cell's selectivity value to a distribution with shuffled trial labels (significance threshold of $p < 0.01$). (**H**) In each area, trial-type decoding accuracy using activity of subsampled neurons is compared in the simple versus the switching task (Rule A trials only). Shading indicates mean ± SEM across sessions. p value is for the task factor from a two-way ANOVA (factors: task and neuron number).

The online version of this article includes the following figure supplement(s) for figure 4:

**Figure supplement 1.** Task training times, task performance, and neuronal trial-type selectivity sequences across task conditions.

**Figure supplement 2.** Trial-type selectivity for sessions with similar running patterns.

**Figure supplement 3.** Neuronal activity and selectivity in Rule B of the switching task compared to the simple task.

**Figure supplement 4.** Neuronal selectivity in the switching task across rules.

In summary, we found increased activity and selectivity levels in RSC and PPC, but not V1, when comparing mice performing the switching task to mice performing the simple task. The relative increase was observed in both rules constituting the switching task, supported by mixed selectivity in single neurons. Thus, task complexity affected not only the necessity of cortical association areas for task performance as revealed in our photoinhibition experiments, but also their activity and selectivity levels.

## Previous switching task experience increases neural trial-type selectivity in PPC and RSC

We then examined if previous experience in the switching task affected the neural activity patterns in the simple task. Similar to our tests of cortical necessity, we compared neural activity during the simple task in mice that either had or had not been trained previously in the switching task (*Figure 5A*, *Figure 4—figure supplement 1*). We trained one group of mice on the switching task and then permanently transitioned these mice to the simple task. A separate set of mice was trained only on the simple task. We thus compared the activity patterns in PPC, RSC, and V1 in mice in the same task except with distinct experience. As previously observed in our photoinhibition experiments, both groups of mice performed the simple task at similarly high levels irrespective of previous task experience (*Figure 4—figure supplement 1*).

Strikingly, during the simple task, neurons in RSC and PPC had higher activity in mice with experience in the switching task than in mice trained only in the simple task (*Figure 5B*). Furthermore, in mice with switching task experience, RSC and PPC neurons had higher average selectivity for the trial type and higher fractions of neurons with significant trial-type selectivity (*Figure 5C, D*). As a result, the decoding of the trial type from population activity was more accurate in these areas in mice with the complex-task experience (*Figure 5E*). Notably, selectivity in V1 neurons was similar between mice with and without complex-task experience. Therefore, the activity patterns of single neurons in PPC and RSC, including mean activity levels and selectivity, are strongly influenced not only by the current task performed, but also by previous task experience, mirroring the effect of previous task experience on the necessity of these areas for task performance.

## Switching task experience decreases noise correlations

A key feature of neural codes beyond the properties of single cells is the collective activity of populations of neurons. Properties of population codes affect the amount of information in neural populations and have been shown to depend on behavioral context, task learning, and other factors (*Cohen*

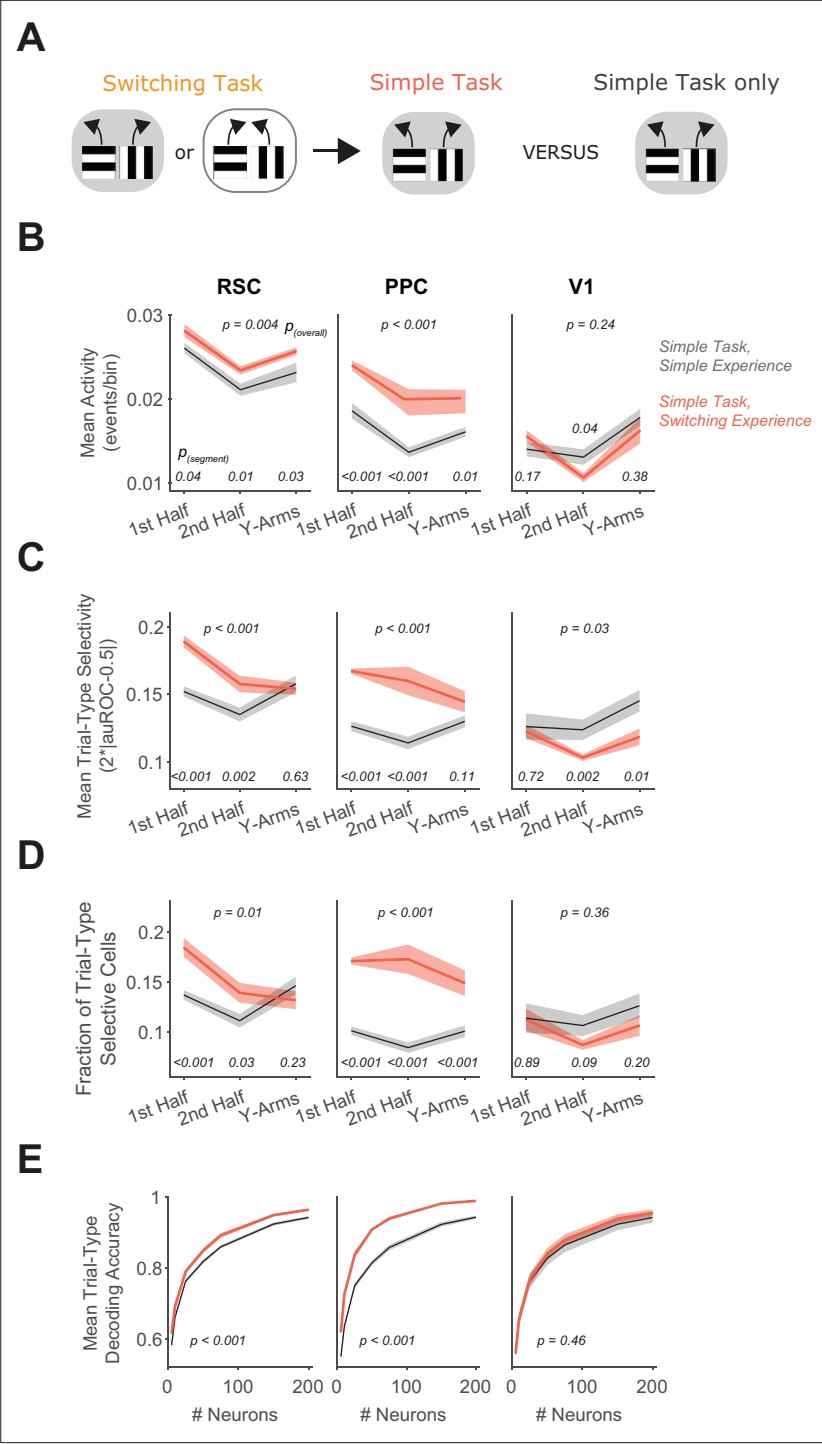

**Figure 5.** Previous switching task experience increases trial-type selectivity in RSC and PPC. (**A**) Schematic of the training history sequence. One group of mice was trained on the switching task and then permanently transitioned to the simple task. This group of mice was compared to another group trained only on the simple task. (**B**) In each area, mean activity levels across cells by maze segment are compared in the simple task in mice with (red) and without (gray) previous experience in the switching task. Shading indicates mean ± standard error of the mean (SEM) of bootstrapped distributions of the mean. $p_{(segment)}$ shows p values of two-tailed comparisons of bootstrapped distributions per maze segment. $p_{(overall)}$ shows the p value for the previous task experience factor from a two-way analysis of variance (ANOVA; factors: previous task experience and maze segment). Simple task: n = 3 mice, 4 sessions per mouse, cells per session by area: RSC: 1438 ± 217, PPC: 456 ± 172, V1: 498 ± 170 (same

*Figure 5 continued on next page*

*Figure 5 continued*

dataset as in *Figure 4D–H*). Simple task after switching task experience: *n* = 2 mice, 3 and 5 sessions per mouse, neurons per session by area: RSC: 1407 ± 327, PPC: 744 ± 219, V1: 351 ± 90 (mean ± standard deviation [SD]). (**C**) Similar to (**B**), except for the metric of trial-type selectivity, that is 2\*|auROC − 0.5|. (**D**) Similar to (**B**), except for the fraction of trial-type selective cells as determined from comparing each cell's selectivity value to a distribution with shuffled trial labels. (**E**) In each area, trial-type decoding accuracy using activity of subsampled neurons is compared in the simple task in mice with and without previous experience in the switching task. Shading indicates mean ± SEM across sessions. p value is for the previous task experience factor from a two-way ANOVA (factors: previous task experience and neuron number).

*and Kohn, 2011*). We thus examined if features of the neural population code are also affected by task complexity and past experience. We took advantage of the simultaneous imaging of hundreds of neurons per area and analyzed the correlation in activity for pairs of neurons on a given trial type, a measure commonly referred to as noise correlation, that quantifies trial-to-trial co-fluctuations in neurons (*Cohen and Kohn, 2011*). As for analyses of trial-type selectivity, we restricted analyses to the maze traversal period and only included correct trials from high-performance periods (Methods). The noise correlations within individual areas and across pairs of areas were lower on average in mice performing the switching task than in mice performing the simple task (*Figure 6A–C*, *Figure 4— figure supplement 3*). Notably, noise correlations were not only lower in RSC and PPC, but also within V1 and in V1 interactions with RSC and PPC, suggesting that changes in noise correlations may be a more global phenomenon than changes in selectivity. Strikingly, when we compared mice with different experience as they performed the same simple task, we also observed a difference in noise correlations both within and across cortical areas, with lower correlations in mice with switching task experience compared to mice trained only in the simple task (*Figure 6A–C*). Therefore, mice with different task experience have significant differences in their population codes as they perform the same task.

Noise correlations can in some cases limit the information contained in a neural population because these correlations are co-fluctuations in activity that cannot be removed by averaging across neurons (*Averbeck et al., 2006*; *Kafashan et al., 2021*; *Panzeri et al., 1999*; *Zohary et al., 1994*). To reveal the impact of correlations on coding in our experiments, we disrupted noise correlations by shuffling trials of a given trial type separately for each neuron and repeated the trial-type decoding. The accuracy of decoding the trial type was slightly higher with correlations disrupted (*Figure 6D*). Therefore, the lower correlations in mice performing the switching task or the simple task with switching task experience boosts information encoding along with higher trial-type selectivity levels.

Thus, these results reveal that the activity patterns in single neurons and neural populations are shaped by previous task experience. Together, our findings demonstrate that different sets of cortical areas and distinct neural activity patterns are utilized for the same task depending on an individual's training history.

## Discussion

Our central finding is that mice with enhanced cognitive experience due to previous training on complex tasks require PPC and RSC to perform a simple task, whereas in mice without this previous training, PPC and RSC are largely dispensable for performing the same, simple task. Thus, the necessity of cortical areas depends on factors separate from the task itself. Here, we aimed to vary the 'cognitive experience' of the mouse, which we define broadly as past learning of task rules and cue–action associations. We varied the cognitive experience by adding delay periods (delay task) or frequent switches of associations within a session (switching task) to a simple navigation task. This resulted in tasks of varying complexity, but the maze shape, and thus behavioral outputs needed, and choice-informative cues were identical between the complex and simple tasks. Therefore, the differential cortical involvement in the groups of mice with and without training on the complex tasks is likely due to cognitive experience instead of sensory or motor learning.

Studies of perceptual experience and motor learning have emphasized that cortical necessity for task performance decreases with experience (*Chowdhury and DeAngelis, 2008*; *Hwang et al., 2019*; *Kawai et al., 2015*) but see *Liu and Pack, 2017*. Instead, we found an increased necessity

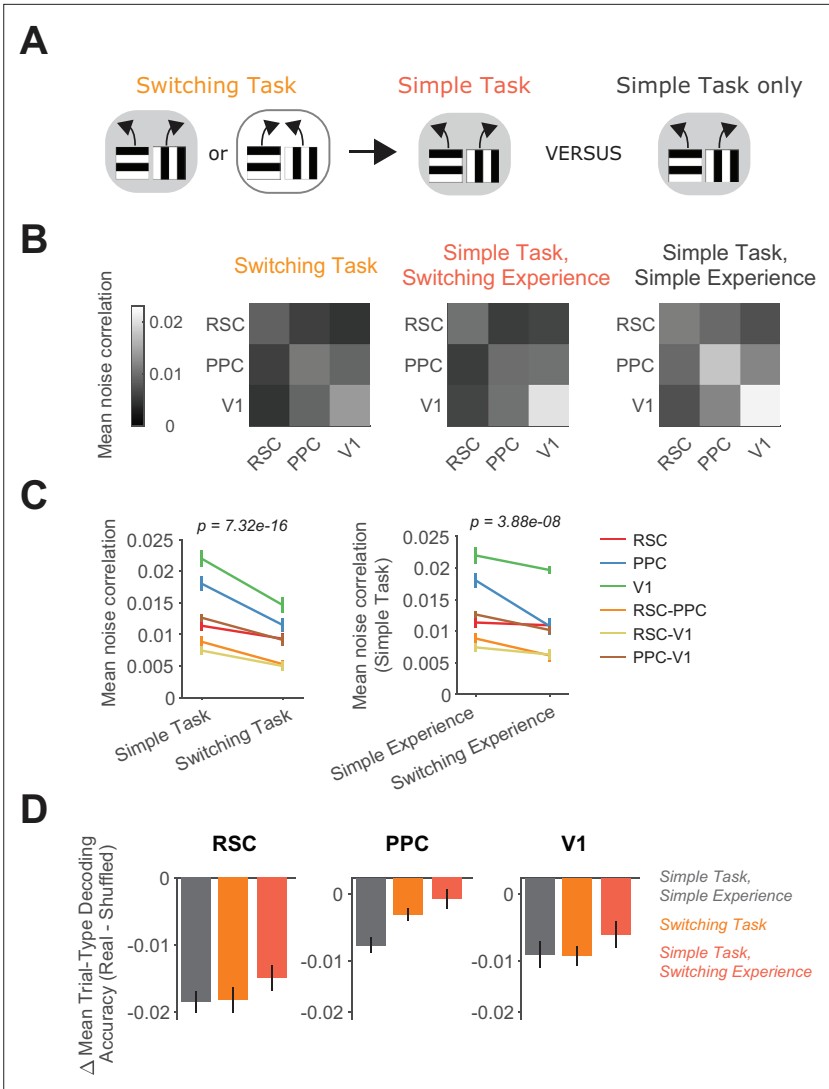

**Figure 6.** Switching task experience decreases noise correlations. (**A**) Schematic of the training history sequence. One group of mice was first trained on the switching task and then permanently transitioned to the simple task. Another group was trained only on the simple task. (**B**) Mean pairwise noise correlations within and across areas from bootstrapped distributions of the mean in the switching task (left), the simple task with previous switching task experience (middle), and the simple task with only simple task experience (right). Noise correlations were calculated on spatially binned data in correct trials during high-performance periods (Methods). (**C**) Left: Comparison of mean noise correlations in the switching task versus the simple task, p value is for the task factor from a two-way analysis of variance (ANOVA; factors: task and area combination). Error bars indicate mean ± standard error of the mean (SEM) across sessions per area combination (n = 6 area combinations, n sessions: 12 (simple task) and 12 (switching task)). Right: Similar to left, except for the comparison of simple task noise correlations with and without previous switching task experience. n sessions: 12 (simple experience) and 8 (switching experience). (**D**) For each area and task or previous task experience, the difference in trial-type decoding accuracy between neural populations (200 subsampled neurons) with intact and disrupted noise correlations. Noise correlations were disrupted by shuffling trials independently for each cell within a given trial type. Error bars show mean ± SEM across sessions.

of cortical association areas due to cognitive experience in two distinct, complex tasks (delay and switching tasks). Future work should investigate how differences in the types of experience (cognitive versus perceptual), tasks, and areas studied (association versus sensory or motor cortices) influence whether experience increases or decreases cortical necessity. In addition, these earlier works did not identify differences in neural tuning in MT with perceptual experience despite differences

in necessity, leading to the proposal that differences in cortical necessity with experience arise from whether an area's activity is read out by a downstream network. On the other hand, studies comparing various behavioral tasks with distinct levels of cortical necessity reported differences in neural activity across tasks (*Harvey et al., 2012*; *Pinto et al., 2019*), suggesting that differences in cortical necessity may result from differential neural activity in the same areas. Here, we did find differences in neural activity depending not only on current task complexity, but also on training history. During the same task, mice with complex training experience had higher selectivity in single neurons and weakened neuron–neuron correlations, compared to mice without this experience, which together allowed for easier decoding of the relevant information from a population of neurons. We found these increases in neural selectivity specifically in RSC and PPC, that is the same areas for which we observed differential necessity based on training history. However, these areas already contained high amounts of task-relevant information in the simple task without complex-task experience, contrasting the small performance deficits from inhibiting these areas in the same condition. This picture is in line with several studies reporting encoding of task variables in brain areas that leave task performance largely intact when inhibited (*Allen et al., 2017*; *Katz et al., 2016*; *Zatka-Haas et al., 2021*). Therefore, although the boost in neural selectivity with complex-task experience increases the encoded information, it appears unlikely to be the sole reason for the large increase in cortical necessity for task performance. Rather, increased cortical necessity may result from a combination of increases in task information, reshaped representations of information, and/or modifications to information readouts (*Ruff and Cohen, 2019*). Further work will be needed to test directly the relationship between specific features of the neural code and the causal roles of PPC and RSC in decision tasks.

We found that cortical association areas had higher selectivity in their neural activity and were more strongly required for the complex tasks than the simple task. This finding supports the notion that cortex is needed for cognitively more challenging tasks. Previous work presented a similar finding but focused on tasks with a wide gap in their demands, such as comparing running toward a visual target versus using learned cue–choice associations to make navigation decisions (*Buschman et al., 2011*; *Ceballo et al., 2019*; *Fuster, 1997*; *Harvey et al., 2012*; *Lashley, 1931*; *Pinto et al., 2019*; *Sarma et al., 2016*). In our work, we extended this concept by keeping the visual and behavioral aspects of the complex and simple tasks as similar as possible and adding specific cognitive challenges. We added task components requiring short-term memory on various timescales, which is hypothesized to cause increased PPC necessity for task performance based on a synthesis of previous PPC inactivation studies (*Lyamzin and Benucci, 2019*). Indeed, the insertion of a delay period or of frequent association switches caused large increases in PPC necessity. However, PPC necessity was not restricted to the task parameters changed relative to the simple task. In the delay task, PPC was necessary not only in the delay period but also in the cue period, possibly due to additional top-down attentional signals in the delay task relative to the simple task (*Freedman and Ibos, 2018*). In the switching task, PPC activity was dispensable in the intertrial interval following rule switches, when previous cues, choices, and outcomes are likely combined to update rule beliefs. Thus, PPC may be especially important not for the combination of sensory cues with internal signals alone, but for their transformation into actions, consistent with strong encoding of ongoing movement in this area (*Minderer et al., 2019*).

In contrast to PPC, inhibition of RSC had an intermediate effect on performance even in the simple task, in line with previous work on RSC supporting spatial navigation in general by relating egocentric sensory cues to allocentric positional information (*Alexander et al., 2020*; *Alexander and Nitz, 2015*; *Fischer et al., 2020*). However, like for PPC, inhibiting RSC caused larger deficits when navigation was embedded in complex tasks. In those cases, because the navigation and visual components of a trial were identical across tasks, larger RSC involvement was unlikely due to increased difficulty of navigating the visual scenery itself. These results support the emerging view that RSC plays a crucial role in cognitive aspects of navigation decisions, integrating visual cues flexibly with memory signals to plan navigational routes or choices (*Franco and Goard, 2021*; *Spiers and Maguire, 2006*; *Stacho and Manahan-Vaughan, 2022*). The finding of highly selective activity levels in single neurons in the switching task, especially in RSC, further highlights this point. The imaging data we collected during the switching task provide a starting point for future analysis of how PPC and RSC contribute to various aspects of goal-directed navigation.

The long-lasting effect of cognitive experience on the necessity of PPC and RSC has important implications for the experimental study of cortical involvement in decision tasks. Many studies,

including our previous work, test an area's involvement or activity patterns in a single task and develop interpretations of an area's functions by extrapolating across studies (*Lyamzin and Benucci, 2019*). However, given that factors beyond the task-of-interest contribute to an area's necessity and activity in a task, we encourage consideration of a variety of factors that have often been ignored and not reported. Because prior task expertise can have a large effect on the involvement of cortical areas, it seems critical to report the full details of how animals were trained on a task and what previous experiences they encountered, both of which are commonly omitted from publications.

What may be the reason for persistent cortical necessity from cognitive experience? Increased PPC and RSC involvement relative to mice lacking complex-task experience is unlikely to stem from differences in task exposure times. The modest performance deficits from cortical inhibition in the simple task for mice without complex-task experience were stable over long-time periods, and our training and task exposure times were similar to or exceeded those in other studies reporting substantial RSC involvement and task-related RSC activity (*Fischer et al., 2020*; *Franco and Goard, 2021*; *Milczarek et al., 2018*; *Smith et al., 2018*; *Vedder et al., 2017*). Importantly, our experimental design meant that for mice with complex-task experience, the simple task did not present an unfamiliar challenge but was a simplification of a well-known task. Mice trained on the delay task transitioned to the task they originally learned before the delay was introduced in their training protocol, and mice expert at the switching task were simply transitioned to a setting in which they did not experience rule switches anymore. Simple task performance on trials without cortical silencing was similarly high in mice with and without complex-task experience, suggesting that all groups of mice found the simple task similarly easy. In contrast to comparisons across tasks of varying complexity, this represents an interesting dissociation between cortical necessity and apparent task difficulty. We speculate that learning of the cue–choice associations in the complex tasks may create a network that is then used for the same associations in the simple task, with PPC and RSC as essential nodes that cannot be readily disengaged even when task complexity decreases. This reliance on association areas for simple task performance may equip animals with the capacity to quickly adjust behaviorally should the task complexity increase again. This idea is supported by the fact that mice previously trained on the switching task could still perform the switching task after 2 weeks of only simple task exposure.

Previous work has investigated how prior learning impacts subsequent learning, including in studies describing schema learning, learning sets, learning-to-learn meta-learning, and transfer learning. It will be of interest to investigate if these types of learning could contribute to our findings. However, we note that in our case, the cue–choice associations in the simple task were identical to those learned previously in the complex tasks for mice with experience in the delay or switching tasks. Thus, it is unlikely that mice had to undergo substantial learning to perform the simple task after transitioning from a complex task, which was supported by high behavioral performance in even the first session in the simple task after this transition.

Together, our results indicate that the cortical implementation of a task flexibly depends on initial conditions, as defined by past experience, and the overall optimization goal for the animal, which in many cases is not just a single task but also previous tasks or other tasks occurring in parallel (*Golub et al., 2018*; *Sadtler et al., 2014*). These results highlight the tremendous flexibility of the brain to perform outwardly identical tasks using distinct sets of brain areas and neural activity patterns and raise exciting challenges for understanding neural computation in the framework of dynamic and distinct neural solutions for a given cognitive problem. We propose that understanding cognitive processes will require considering the wider set of functions an animal is trying to optimize, beyond the computation of interest in a particular study. To understand how long-ago cognitive experience and current cognitive demands set up these different neural circuit landscapes for outwardly identical navigation decisions, we suggest to carefully control for and to intentionally vary cognitive experience in laboratory settings (*Plitt and Giocomo, 2021*; *Sharpe et al., 2021*), thereby approximating more naturalistic scenarios. We anticipate this approach will be particularly powerful to illuminate neural circuit differences underlying interindividual variability and changes in neural dynamics during learning (*Oby et al., 2019*; *Sadtler et al., 2014*).

# Materials and methods

## Mice

All experimental procedures were approved by the Harvard Medical School Institutional Animal Care and Use Committee and were performed in compliance with the Guide for the Care and Use of Laboratory Animals. All optogenetic inhibition data were acquired from 19 male VGAT-ChR2-YFP mice (The Jackson Laboratory, stock 014548). All calcium imaging data were acquired from six C57BL/6J-Tg (Thy1-GCaMP6s) GP4.3Dkim/J mice (stock 024275) of both sexes (five females, one male). Mice were 11–32 weeks old at the start of behavioral training. Age at training start did not vary systematically across tasks (simple task: 21 ± 7 weeks ($n$ = 7), switching task: 17 ± 5 weeks ($n$ = 13), delay task: 21 ± 11 weeks ($n$ = 5), mean ± standard deviation (SD), including photoinhibition and calcium imaging mice). Mice were kept on a reverse dark/light cycle and housed in groups of 2–3 littermates per cage (mice for optogenetic inhibition) or single housed (mice for calcium imaging).

## Virtual reality setup

Virtual reality environments (*Harvey et al., 2009*) were designed and operated in VirRMEn (Virtual Reality Mouse Engine) (*Aronov and Tank, 2014*). A novel, custom-made compact virtual reality design was employed (overall dimensions of approximately 15 inches wide × 21 deep × 18 high), using modifiable laser-cut acrylic and mirror pieces. A micro projector (Laser Beam Pro) projected the virtual environment onto a double-mirror system and a 15-inch diameter half-cylindrical screen. The mouse was head-fixed on top of an 8-inch diameter Styrofoam spherical treadmill. The mouse's position in the virtual environment, and thus the projection, was controlled by the mouse's movement of the treadmill, which was measured with two optical sensors (ADNS-9800, Avago Technologies) placed 90° apart from each other beneath the ball. The treadmill velocity was translated into pitch, roll, and yaw velocity relative to the mouse's body axis using custom code on a Teensy microcontroller (PJRC). Pitch controlled forward/backward movement in the virtual world, while roll controlled lateral movement. The virtual view angle was fixed so that the mouse could not rotate in the virtual world. Designs for the virtual reality apparatus are available at https://github.com/HarveyLab/mouseVR.

## Behavioral training

Mice were limited to 1 ml of water per day for several days before starting training. Body weight and body condition were checked daily. Mice were maintained at approximately 80% of their body weight prior to water restriction and received additional water if their body weight fell below 75% of their original weight. Mice were trained daily for 45–80 min (except for some weekends, when they received 1 ml of water without training). In the first training phase, to get accustomed to the experimental setup and to moving in virtual space, mice had to run down the length of a rectangular environment ('linear maze') toward a checkerboard pattern (*Figure 1—figure supplement 1*). When reaching the checkerboard, they received a reward (3–4 µl of 10% sweetened condensed milk in water delivered through a lick spout) and were teleported back to the beginning after a brief intertrial interval (ITI, 1 s). The linear maze contained the visual cues on the walls that the mice would later learn to associate with rewarded choice directions. Linear maze length was increased on a session-by-session basis until mice completed approximately 200 trials of a 200 cm maze in 60 min (minimum/maximum maze length of 10/200 cm). Linear maze training typically lasted 7–14 sessions.

### Y-maze training, general procedures

After training on the linear maze, mice were transitioned to a Y-shaped maze (180 cm long) in which they had to run toward one of two possible Y-arms to get rewarded. In all tasks, visual cues presented on maze walls were associated with rewarded choice arms at the end of the maze. Initially, in all tasks, the correct choice was signaled with a checkerboard at the end of the correct Y-arm (100% "visually guided trials", *Figure 1—figure supplement 1*). Throughout training, the fraction of "visually guided trials" was gradually reduced on a session-by-session basis by the experimenter, based on the mouse's previous performance. For the simple and delay task, average performance in the preceding session had to exceed 80%. For the switching task, overall performance including periods after rule switches had to exceed 70% correct, performance after rule switches was inspected for drops followed by recovery over tens of trials, and mice had to obtain at least two rule switches per session. Across all

tasks, after extensive training, a minority of trials were still 'visually guided trials' (10–15%). In nonvi-sually guided trials, the checkerboard appeared for 2 s as visual feedback once the mouse made a correct choice, followed by the reward and an ITI with a gray screen of 2 s. After incorrect choices, no checkerboard was presented, and the ITI lasted for 4 s. A different cohort of mice was trained on each task unless specified otherwise. As some mice developed biases during training, making predomi-nantly left or right choices, we employed a bias correction algorithm in some training sessions. If the mouse made the same choice on the last five trials irrespective of correctness, the next trial would contain a maze in which the opposite choice would be correct. This bias correction was only used at intermediate training stages and was not employed during inhibition or calcium imaging sessions.

## Simple task

In the simple task, mice encountered one of two possible cues in a given trial, with a fixed association between cue identity and rewarded Y-arm. Horizontal gratings were associated with a left rewarded choice and vertical gratings were associated with a right rewarded choice. This pair of associations constitutes what we call 'Rule A' in the switching task. Visual cues were present along the entire extent of the maze, including the stem and the Y-arms.

## Delay task

Mice trained in the delay task were first trained in the simple task (*Figure 1—figure supplement 1*). After reaching high-performance levels in the simple task (at least 90% correct), a delay was intro-duced, that is a neutral visual texture present in all trial types that was uninformative about the choice to make on a given trial. In the first sessions of delay task training, the delay texture was only present in the Y-arms of the maze. Then the delay onset (i.e., cue offset) was gradually shifted earlier in the trial by 10 cm increments on a session-by-session basis if performance in the preceding session exceeded 80% correct, until only the first half of the Y-maze stem contained the visual cue (50 cm). Thus, mice had to traverse the rest of the maze without the informative cue on the walls. Once a 50-cm delay length was reached, the spatial delay length was kept constant across sessions including the photoin-hibition sessions, and mice spent $4.06 \pm 0.87$ s in the delay period overall (second half of maze stem + maze funnel), of which $1.4 \pm 0.35$ s were spent in the delay period of the maze stem only (mean $\pm$ SD, $n = 85$ sessions across 7 mice). A subset of mice (two out of seven: Mouse IDs 38, 41) was used for photoinhibition experiments both during the simple task before delay task training, and later during the delay task after delay task training.

## Switching task

In the switching task, mice encountered the trial types that were visually identical to those in the simple task, but now the associations between visual cue and rewarded choice were switched in blocks (Rules A and B). In 'Rule B', mice had to make the opposite choices given the same cue identi-ties as in Rule A to get rewarded, that is horizontal/vertical gratings were associated with a rightward/ leftward choice, respectively. Rule switches were not explicitly signaled to the mice, so they had to integrate information of past cues, choices, and rewards to inform their belief about the current rule. Rule switches were present from the first day of the Y-maze training period. Given the increased cogni-tive demand of the switching task, the fraction of visually guided trials was reduced more slowly in the switching task than in the simple task, resulting in more training sessions in the switching task than the simple task (*Figure 1—figure supplement 1*). The initial rule in a given session was alternated on a daily basis, starting out with Rule A on the first day of training. Within a training session, a rule switch occurred if several criteria were met: a minimum of 75 trials from the previous rule switch or the session start, a minimum average performance in the last 30 trials of 85% correct, and a correct choice on the immediately preceding trial. Mice encountered 2–3 rule switches per session, indicating they could repeatedly switch associations successfully.

## Run-to-visual-target task

To establish a baseline for cortical involvement in a simple navigation task in which mice did not use visual cues on the maze walls to guide their choices, we employed a run-to-visual target task. The maze had the standard Y-shape architecture, but no informative visual cues on the maze walls. Instead, mice simply had to run toward the checkerboard present at one of the two Y-arm ends in each trial.

The checkerboard location (left or right) was randomly chosen on each trial. We used mice previously trained in either the simple task only (*Figure 1—figure supplement 4*) or trained on a complex task (switching or delay) before simple-task-only exposure (*Figure 3—figure supplement 1*), so the mice already knew that the checkerboard signified a reward location and required minimal training on this task (1–2 days prior to photoinhibition).

## Photoinhibition experiments

### Clear skull cap surgery

We followed procedures described previously (*Guo et al., 2014*; *Minderer et al., 2019*). In brief, the scalp and the periosteum were removed from the dorsal skull surface. The skull surface was covered with a thin layer of cyanoacrylate glue (Insta-Cure, Bob Smith Industries). A bar-shaped titanium head-plate was attached to the interparietal bone using dental cement (Metabond, Parkell). Several layers of transparent dental acrylic (Jet Repair Acrylic, Lang Dental, P/N 1223-clear) were applied to the parietal and frontal bones to create a transparent skull cap. In a subsequent procedure preceding photoinhibition experiments, the acrylic was polished with a polishing drill (Model 6100, Vogue Professional) with denture polishing bits (HP0412, AZDENT). Clear nail polish was applied on top of the polished acrylic (Electron Microscopy Sciences, 72180). An aluminum ring was attached to the skull using dental cement mixed with carbon powder (Sigma-Aldrich) for light shielding.

### Experimental setup and logic

Light from a 470-nm collimated laser (LRD-0470-PFFD-00200, Laserglow Technologies) was focused onto the skull using an achromatic doublet lens ($f$ = 300 mm, AC508-300-A-ML, Thorlabs). We coupled the laser to a pair of galvanometric scan mirrors (6210H, Cambridge Technology) in combination with rapid analog laser power modulation to allow fast movement of the focused beam between cortical target sites. At the focus, the laser beam had a diameter of approximately 200 μm.

We started photoinhibition only after mice reached expert performance in a given task (criterion for the simple or delay task: performance of approximately 85% correct or higher, criteria in the switching task: at least two rule switches with only 15% visually guided trials per session). We thus started inhibition after shorter training times in the simple task group of mice, compared to the complex task (delay or switching) groups that required longer training times to reach expert performance (see *Figure 1*). In each session, we bilaterally targeted PPC, RSC, S1, and a control site outside of the brain (on the dental cement) on separate, interleaved trials. For PPC, S1, and control targets, we used single bilateral laser spots, with laser power sinusoidally modulated at 40 Hz and a time-average power of approximately 6.5 mW/spot. For RSC, we used three spots on each hemisphere to match the region's anatomical extent, with laser power sinusoidally modulated at 20 Hz and a mean power of approximately 5 mW/spot. The target coordinates in mm from bregma were: RSC (−3.5, −2.5, −1.5 anterior–posterior [AP]; 0.5 medial–lateral [ML]); PPC (−2 AP, 1.75 ML); S1 (−0.5 AP, 2.5 ML); control (2 AP, 5 ML). Based on previous calibration studies (*Guo et al., 2014*; *Pinto et al., 2019*), we estimate that the laser powers employed here inhibited a cortical area with a radius of 1–2 mm per inhibition spot and a depth spanning the entire cortex (900 μm), indicating inhibition of both dysgranular and granular areas of RSC.

In each experimental session, blocks of at least 50 trials without laser light were alternated with laser-on blocks of 50 trials. Laser-on blocks only started if the mouse's average performance in the preceding 30 trials was at least 85% correct. Thus, in the switching task, laser-on blocks occurred once the mouse had reached stable performance in the current rule block. In the switching task, rule switches happened after the end of each laser-on block. Within laser-on blocks, approximately 50% of trials were control trials, and the laser target location was randomly chosen for each trial. Within a trial, the laser was on from 0.5 s before visual cue onset at the trial beginning until the mouse reached the end of the maze, excluding the visual feedback and reward/ITI periods of the trial. In the run-to-visual-target task and a subset of sessions in the simple task, a single long block of 200 laser-on trials was delivered after the mouse reached high-performance levels, again with approximately 50% control trials randomly interleaved with cortical inhibition trials. In the simple task, inhibition effects did not vary between sessions with laser-on blocks of 50 or 200 trials.

For experiments with maze segment-specific inhibition in the delay task (*Figure 1—figure supplement 5*), the stem of the Y-maze was doubled in length to 200 cm, and the laser-on period per session

was restricted to either only the cue period (maze beginning until delay onset) or the delay period (delay onset until the end of the maze, excluding visual feedback and reward/ITI periods). Cue only and delay only inhibition sessions were generally alternated from day to day. Mice spent 4.99 ± 1.12 s in the delay period in these experiments including the Y-arms of the maze, of which 2.26 ± 0.53 were spent in the delay period of the maze stem only (mean ± SD, 71 sessions across 5 mice).

For experiments with ITI inhibition in the switching task (*Figure 1—figure supplement 6*), PPC was the only cortical inhibition target. PPC was inhibited during the ITI for 50 consecutive trials following either the first or the second rule switch per session. In all other trials, the laser was steered to the control location during the ITI so that rule switches could not be inferred simply from the presence of laser light. Inhibition started upon the mouse reaching one of the two possible Y-arm maze ends and lasted throughout checkerboard feedback presentation and the ITI/reward delivery.

## Long-term experimental stability and order of experiments across tasks

To ensure stability of experimental conditions across long times, we maintained constant laser power by measuring maximum power daily and cleaning optics if necessary. To ensure stability of inhibition conditions per mouse, we verified the alignment of the laser beam orientation to the mouse's skull by creating a laser cross pattern to be centered on Bregma and to be aligned with the mouse's AP–ML skull axes. To aid in the latter, we added marks on the mouse's dental cement in AP and ML that the laser cross had to intersect. To change alignment horizontally or vertically, we moved an X–Y stage that the laser apparatus was mounted on. We controlled for rotation by slight adjustments to the posts holding the mouse's headplate. Experiments in different cohorts across tasks were not systematically interleaved but were clustered in time as we iterated through hypotheses throughout the project. We have several indicators that experimental conditions remained stable and that differences across tasks were not the result of experimental drift. First, data for the switching task were collected in several groups of mice spanning the full range of data collection times, yet the average inhibition effects on performance were similar (group 1 [early] (mouse IDs 24, 27, 42) versus group 2 [late] (mouse IDs 72, 73, 74): ΔFraction correct for Rule A: S1: −9 ± 2% versus −7 ± 2%, RSC: −31 ± 1% versus −38 ± 4%, PPC: −34 ± 3% versus −30 ± 3 %, mean ± SEM). Second, data from the simple task with notably smaller inhibition effects were collected in between these two groups and were partly on overlapping days as the first group. Third, in some mice (Mouse IDs 38, 41), we tested the effect of inhibition in the identical mice in the simple task first, as well as after they learned the delay task, observing large differences in cortical inhibition effects in the same mice within weeks (ΔFraction correct in simple versus delay task: S1: −1 ± 2% versus −18 ± 18%, RSC: −16 ± 6% versus −36 ± 3%, PPC: −5 ± 2% versus −37 ± 8%, mean ± SEM).

# Calcium imaging experiments

## Large chronic cranial window surgery

We slightly modified procedures described previously (*Kim et al., 2016*; *Kilic et al., 2020*). Mice were injected with dexamethasone (3 µg per g body weight) 4–8 hr prior to surgery and anesthetized with isoflurane (1–2% in air). A cranial window surgery was performed to either fit a 'crystal skull' curved window (LabMaker UG) exposing the dorsal surface of both hemispheres (*Kim et al., 2016*) or the left hemisphere only (*Kilic et al., 2020*), or to fit a stack of custom laser-cut quartz glass cover-slips (three coverslips with #1 thickness each [Electron Microscopy Sciences], cut to a 'D'-shape with maximum dimensions of 5.5 mm medial–lateral and 7.7 mm anterior–posterior, and glued together with UV-curable optical adhesive [Norland Optics NOA65]), exposing the left cortical hemisphere. The skull was kept moist using saline throughout the drilling procedure and soaked in saline for 1–2 min before being lifted. The dura was removed before sealing the window using dental cement (Parkell). A custom titanium headplate was affixed to the skull using dental cement mixed with carbon powder (Sigma-Aldrich) to prevent light contamination. A custom aluminum ring was affixed on top of the headplate using dental cement. During imaging, this ring interfaced with a black rubber balloon enclosing the microscope objective for light shielding.

## Calcium imaging setup and data acquisition

Data were collected using a large field of view two-photon microscope assembled as described previously (*Sofroniew et al., 2016*). In brief, the system consisted of a combination of a fast resonant

scan mirror and two large galvanometric scan mirrors allowing for large scan angles. Together with a remote focusing unit to rapidly move the focus depth, this setup enabled random access imaging in a field of view of 5 mm diameter with 1 mm depth. The setup was assembled on a vertically mounted breadboard whose XYZ positions and rotation were controlled electronically via a gantry system (Thorlabs). Thus, to position the imaging objective with respect to the mouse, the position and rotation of the entire microscope were adjusted while the position of the mouse remained fixed. The excitation wavelength was 920 nm, and the average power at the sample was 60–70 mW. The microscope was controlled by ScanImage 2016 (Vidrio Technologies). We imaged in three distinct regions in the left cortical hemisphere: V1, PPC, and RSC. These regions were identified based on retinotopic mapping (see below). In each region, we acquired images in layer 2/3 from two planes spaced 50 µm in depth, at 5.36 Hz per plane at a resolution of 512 × 512 pixels (600 µm × 600 µm). Imaging was performed in expert mice in the simple task, switching task, and simple task after switching task experience (criterion for the simple task: performance of approximately 85% correct or higher, criteria in the switching task: at least two rule switches with only 15% visually guided trials per session). The stem of the Y-maze was extended by 50% (50 cm) compared to the maze architecture in photoinhibition sessions, resulting in a maze length of 230 cm. Each imaging session lasted 45–80 min. During imaging, slow drift of the image was occasionally corrected manually by moving the gantry to align the current image with an image from the beginning of the session. For synchronization of imaging and behavior data, both the imaging and the behavior frame clock were recorded on another computer using Wavesurfer (https://wavesurfer.janelia.org/).

## Retinotopic mapping for selecting calcium imaging locations

We performed retinotopic mapping in mice used for calcium imaging experiments as previously described (*Driscoll et al., 2017*; *Minderer et al., 2019*). Mice were lightly anesthetized with isoflurane (0.7–1.2% in air). A tandem-lens macroscope was used in combination with a CMOS camera to image GCaMP fluorescence at 60 Hz (455 nm excitation, 469 nm emission). A periodic spherically corrected black and white checkered moving bar (*Marshel et al., 2011*) was presented in four movement directions on a gamma-corrected 27 inch IPS LCD monitor (MG279Q, Asus). The monitor was centered in front of the mouse's right eye at an angle of 30° from the mouse's midline. To produce retinotopic maps, we calculated the temporal Fourier transform at each pixel of the imaging data and extracted the phase at the stimulus frequency (*Kalatsky and Stryker, 2003*). These phase images were smoothed with a Gaussian filter (25 µm s.d.). Field sign maps were generated by computing the sine of the angle between the gradients of the average horizontal and vertical retinotopic maps.

For each retinotopic mapping session, we acquired an image of the superficial brain vasculature pattern under the same field of view. We acquired a similar brain vasculature image under the large field of view two-photon microscope. These two reference images were manually aligned and used to directly locate V1 and PPC locations for two-photon imaging. V1 imaging was performed in the most anterior part of V1 closest to PPC. The location for RSC imaging was positioned adjacent to the midline and about 300 µm anterior of the PPC location, thus targeting an anterior portion of dysgranular RSC.

## General analyses

Statistical estimates and significance were generally generated with hierarchical bootstrapping (*Saravanan et al., 2019*), and data are reported as mean ± SEM of hierarchical bootstrap distributions, unless noted otherwise. SEM was calculated as the standard deviation of the means from a bootstrap distribution ($n = 1000$ resampled datasets). For analyses of optogenetic inhibition effects, resampled datasets were generated by sampling with replacement first at the level of sessions pooled across mice and then at the level of trials. For analyses of calcium imaging data, resampled datasets were generated by resampling at the level of sessions then neurons. For significance testing of differences between bootstrap distributions, the probability that one was greater or less than the other, whichever was smaller, was computed. To obtain a p value for a two-tailed test with $\alpha = 0.05$, this probability was doubled. Analyses were performed with custom code in MATLAB. No statistical methods were used to predetermine sample sizes, but our sample sizes were similar to ones in previous publications in the field. Allocation of individual mice into experimental groups, that is behavioral tasks, was not randomized, and co-housed mice were trained on the same behavioral task

and task sequence. Data collection was not performed blind to the experimental groups. Blinding experimenters would have been challenging as experimenters remained present throughout behavioral sessions to ensure the sessions were running smoothly, and many experimental groups were inferable by observing the virtual reality display and rewarded choices over time. Data collection was performed by four different experimentalists. Analyses were also nonblinded but performed by two different experimentalists. A small number of behavioral sessions were excluded from analysis due to low performance of the mouse on control trials. Imaging sessions were excluded in case of noticeable drift after motion correction.

## Analysis of photoinhibition experiments

### Effects of photoinhibition on performance

Performance was quantified as 'fraction correct', the fraction of trials in which the mouse made the correct choice. Chance performance was 50% correct. Effects of cortical inhibition were measured as ΔFraction Correct, the fraction correct with inhibition minus the fraction correct with the laser steered to the control (off-cortex) spot. Fraction correct and ΔFraction Correct were calculated on a session basis. For comparisons of control performance and performance with various cortical inhibition targets within a task, significance levels were adjusted with the Bonferroni method.

### Photoinhibition effects on choice biases

Choice biases were calculated per mouse and task (*Figure 1—figure supplement 2*). For each session, a signed choice bias value for each inhibition target was calculated as: (Frac Corr$_{left}$ − Frac Corr$_{right}$)/ (Frac Corr$_{left}$ + Frac Corr$_{right}$). Thus, a signed choice bias of 1 or −1 indicates that the mouse only made left or right choice, respectively.

## Analysis of calcium imaging experiments

### Preprocessing of imaging data

To correct for motion artifacts, custom code was used as described in detail previously (*Chettih and Harvey, 2019*; *Chettih et al., 2019*). In brief, motion correction was implemented as a sum of shifts on three distinct temporal scales: subframe, full-frame, and minutes-to-hour-long warping. After motion correction, regions of interest (ROIs) were extracted with Suite2P (*Pachitariu et al., 2016*). Afterwards, somatic sources were identified with a custom two-layer convolutional network in MATLAB trained on manually annotated labels to classify ROIs as neural somata, processes, or other (*Chettih and Harvey, 2019*). Only somatic sources were used. After identifying individual neurons, average fluorescence in each ROI was computed and converted into a normalized change in fluorescence (Δ*F/F*). We corrected the numerator of the Δ*F/F* calculation for neuropil by subtracting a scaled version of the neuropil signal estimated per neuron during source extraction:

$F_{neuropilCorrected} = F - 0.7*F_{neuropil}$.

The baseline fluorescence of this trace was estimated as the 8th percentile of fluorescence within a 60-s window (baseline$_{neuropilCorrected}$), and subtracted to get the numerator:

$\Delta F = F_{neuropilCorrected} - baseline_{neuropilCorrected}$

We divided this by the baseline (again 8th percentile of 60-s window) of the raw fluorescence signal to get Δ*F/F*. The Δ*F/F* trace per neuron was deconvolved using the constrained AR-1 OASIS method (*Friedrich et al., 2017*). Decay constants were initialized at 2 s and optimized separately for each neuron.

All analyses were performed on deconvolved activity that was spatially binned along the long axis of the maze (5 cm bins). To be able to compare neural activity across tasks, only correct trials from high-performance periods were included (minimum of 80% correct in a window of 10 trials, which excludes periods after rule switches in the switching task). In the switching task, only trials from a single rule (Rule A, i.e., the vertical grating cue/horizontal grating cue requires a right/left choice) were included, unless noted otherwise. Furthermore, for comparisons of trial-type selectivity, noise correlations, or trial-type decoding across tasks, trials were subsampled to the low number of trials per trial type (i.e., horizontal cue/left trial versus vertical cue/right trial) per session in the switching task when considering only high-performance trials for Rule A ($n = 30$ trials per trial type).

### Trial-type selectivity

To quantify if activity of single neurons was informative about the current trial type, the auROC was calculated for each bin and averaged per maze segment (first half of stem, second half of stem, Y-arms). Trial-type selectivity was defined as the unsigned version of the auROC: 2*|auROC – 0.5| (*Najafi et al., 2020*). To identify neurons with significant trial-type selectivity, for each neuron, unsigned auROC values were recomputed 100 times with shuffled trial labels, and the original value was compared to the resulting distribution. Trial-type selectivity was considered significant if the probability of drawing this value from the shuffled distribution was less than 0.01. The fraction of trial-type selective neurons was calculated for each spatial bin and subsequently averaged per maze segment. To analyze selectivity across both rules in the switching task, all four trial types were considered simultaneously to quantify if single-neuron activity per spatial bin was selective to trials with different cues, choices, or rules (*Figure 4—figure supplement 4A*). In these cases, trials were subsampled to 15 per trial type, rendering 60 trials overall to match the total number used in quantifications of trial-type selectivity when considering only Rule A or Rule B trials. To categorize single-neuron selectivity across spatial bins (*Figure 4—figure supplement 4B, C*), a neuron was first categorized as selective within each rule if its selectivity values were significant for three consecutive spatial bins, and the direction of selectivity per rule (left or right choice) was noted. Selectivity per neuron was then compared across rules to categorize each neuron into the following selectivity categories: 'single trial-type' (only selective in one of the two rules), 'cue' (selective in both rules, but to opposite choices), 'choice' (selective in both rules to the same choice), or 'complex' (selective to opposite choices at different spatial positions within a single rule or both rules).

### Trial-type decoding

For each session and area, at each spatial bin, a linear SVM was trained to predict the current trial type (i.e., horizontal cue/left trial versus vertical cue/right trial) using the activity of a subsample of neurons ($n$ = 5, 10, 25, 50, 75, 150, or 200 neurons, activity of each neuron $z$-scored), with 10-fold cross-validation. This procedure was repeated 40 times for populations of 5 or 10 neurons, and 20 times for populations of 25–200 neurons. For each repetition, the decoding accuracy per bin was calculated as the fraction of test trials in which the trial type was predicted correctly. Decoding accuracy was averaged across spatial bins and repetitions per subsampled population per session. To compare trial-type decoding across tasks, a two-way analysis of variance (ANOVA) with factors for task and population size was used.

### Noise correlations

To measure pairwise noise correlations, we calculated the Pearson correlation coefficient for pairs of neurons separately for each trial type, and then averaged the coefficients across trial types. To compare noise correlations across tasks, a two-way ANOVA with factors for task and brain area combination was used. To assess the effect of noise correlations on population information, we disrupted noise correlations by shuffling the order of trials for each neuron independently for each trial type and repeated the trial-type decoding analysis above. We then calculated the difference in decoding accuracy, subtracting the mean accuracy with disrupted noise correlations from the mean accuracy with intact noise correlations, for a given population size and task.

## Analysis of learning times and running

### Quantification of learning times

To compare the number of training sessions necessary to achieve expert performance across tasks (*Figure 1*, *Figure 4—figure supplement 1*), training sessions were counted from the first day on the Y-maze, after training on the linear maze, until both of the following performance criteria were reached per session: maximum of 20% 'visually guided trials' and average fraction correct of at least 70% correct (switching task) or 85% (simple task and delay task). Note that in the switching task, these performance criteria included all trials per session, including trials following rule switches. For the delay task, an additional performance criterion was a delay length of 50 cm, and only mice without prior photoinhibition sessions in the simple task were included (five out of seven delay task mice).

## Choice decoding based on running parameters

To quantify how well a mouse's reported choice could be decoded from its running parameters in a given task, a generalized linear model was fit using as predictors the instantaneous treadmill velocities for all axes (pitch, roll, yaw), and the lateral maze position. Running parameters were spatially binned along the maze's long axis (5 cm bins), and a different model was trained for each bin with 10-fold cross-validation. In photoinhibition experiments, only control trials were used. In calcium imaging experiments, only correct trials from high-performance periods were used (minimum of 80% correct in a window of 10 trials, which excludes periods after rule switches in the switching task), and in each session, trials were subsampled to the low number of trials per trial type in the switching task when considering only high-performance trials for a single rule per session ($n = 30$ trials per trial type). To subselect calcium imaging sessions with similar running patterns to control for differences in running patterns across tasks (*Figure 4—figure supplement 2*), we used a session-wise criterion of average choice decoding accuracy of 85–95% in the maze stem.

## Acknowledgements

We thank Yvette Fisher, Lauren Orefice, and members of the Harvey lab for feedback on the manuscript, Matthias Minderer for optogenetics designs and code, Shih-Yi Tseng for input on behavioral training, Kıvılcım Kılıç for demonstrating a crystal skull cranial window surgery, and the Research Instrumentation Core at Harvard Medical School. This work was supported by grants from the NIH (R01 MH107620, R01 NS089521, R01 NS108410, DP1 MH125776), an NIMH Diversity Supplement, a Louis Perry Jones Postdoctoral Fellowship (CA), Alice and Joseph Brooks Postdoctoral Fellowships (CA, SK), a Mahoney Postdoctoral Fellowship (CA), a Leonard and Isabelle Goldenson Postdoctoral Fellowship (SK), a Uehara Foundation Research Fellowship (SK), a NARSAD Young Investigator Grant (SK), a JSPS Overseas Research Fellowship (SK), an EMBO postdoctoral fellowship (SS), and a Stuart H.Q. & Victoria Quan Fellowship (NLP).

## Additional information

### Funding

| Funder | Grant reference number | Author |
|---|---|---|
| National Institutes of Health | R01 MH107620 | Christopher D Harvey |
| National Institutes of Health | R01 NS089521 | Christopher D Harvey |
| National Institutes of Health | R01 NS108410 | Christopher D Harvey |
| National Institutes of Health | DP1 MH125776 | Christopher D Harvey |
| Louis Perry Jones Postdoctoral Fellowship | | Charlotte Arlt |
| Alice and Joseph Brooks Postdoctoral Fellowship | | Charlotte Arlt |
| Uehara Memorial Foundation | Uehara Foundation Research Fellowship | Shinichiro Kira |
| Leonard and Isabelle Goldenson Postdoctoral Fellowship | | Shinichiro Kira |
| Brain and Behavior Research Foundation | NARSAD Young Investigator Grant | Shinichiro Kira |
| Japan Society for the Promotion of Science | Overseas Research Fellowship | Shinichiro Kira |

| Funder | Grant reference number | Author |
|---|---|---|
| EMBO | Postdoctoral Fellowship | Sofia Soares |
| Mahoney Postdoctoral Fellowship | | Charlotte Arlt |
| Stuart H.Q. & Victoria Quan Fellowship | | Noah L Pettit |

The funders had no role in study design, data collection, and interpretation, or the decision to submit the work for publication.

## Author contributions

Charlotte Arlt, Conceptualization, Formal analysis, Investigation, Methodology, Project administration, Resources, Software, Supervision, Visualization, Writing – original draft, Writing – review and editing; Roberto Barroso-Luque, Investigation, Methodology, Resources, Software, Writing – review and editing; Shinichiro Kira, Methodology, Project administration, Resources, Supervision, Writing – review and editing; Carissa A Bruno, Ningjing Xia, Investigation; Selmaan N Chettih, Sofia Soares, Noah L Pettit, Methodology, Resources, Writing – review and editing; Christopher D Harvey, Conceptualization, Formal analysis, Funding acquisition, Investigation, Methodology, Project administration, Supervision, Visualization, Writing – original draft, Writing – review and editing

## Author ORCIDs

Charlotte Arlt http://orcid.org/0000-0002-1467-6592
Carissa A Bruno http://orcid.org/0000-0002-7126-2185
Christopher D Harvey http://orcid.org/0000-0001-9850-2268

## Ethics

All experimental procedures were approved by the Harvard Medical School Institutional Animal Care and Use Committee (protocol # 00000073-6) and were performed in compliance with the Guide for the Care and Use of Laboratory Animals.

## Decision letter and Author response

Decision letter https://doi.org/10.7554/eLife.76051.sa1
Author response https://doi.org/10.7554/eLife.76051.sa2

# Additional files

## Supplementary files

• Transparent reporting form

## Data availability

Data have been deposited in Dryad with the DOI: https://doi.org/10.5061/dryad.34tmpg4nr.

The following dataset was generated:

| Author(s) | Year | Dataset title | Dataset URL | Database and Identifier |
|---|---|---|---|---|
| Arlt C | 2022 | Cognitive experience alters cortical involvement in goal-directed navigation | https://doi.org/10.5061/dryad.34tmpg4nr | Dryad Digital Repository, 10.5061/dryad.34tmpg4nr |

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
