## [Editor Report]

The paper is of interest to people interested in understanding the neural substrates of learning and how these can be impacted by previous knowledge and training history. It also has relevance for behavioral neuroscientists when considering possible order effects of experiments.

---

## [Decision Letter]

**Decision letter after peer review:**

Thank you for submitting your article "Cognitive experience alters cortical involvement in navigation decisions" for consideration by *eLife*. Your article has been reviewed by 3 peer reviewers, including Mathieu Wolff as the Reviewing Editor and Reviewer #1, and the evaluation has been overseen by Joshua Gold as the Senior Editor.

Essential revisions:

1) The current narrative on training history is the unique angle to discuss the data but there are other alternative interpretations that cannot be ruled out by the present dataset. For example, one could argue that mice can form learning sets during the whole behavioral procedure and this may impact how well they perform in a given task. Alternatively, it is possible that switching from one task to another in the exact same environment is actually challenging behavioral flexibility. These are important points to consider, as all referees expressed specific concerns along these lines and you can find in the individual reviews multiple options and possibilities to address that. It is expected that if no new data is provided, substantial revisions in the introduction, discussion and possibly the title are necessary to have a more balanced analysis of the behavioral data.

2) The calcium imaging dataset does not deliver a very clear message because the link with the functional study is not very clear (see ref 2) and no clear rationale is provided to consider only very specific portions of the data (see ref 1).

3) On the theoretical ground, authors need to better explain why these 3 particular tasks were selected and why they put such a strong emphasis on "decision-making" (see ref 1 and 2). These tasks do not seem to correspond to classic "decision-making" tasks. The take on decision-making is thus unclear and needs to be justified or removed to provide a fair and accurate description of each task.

*Reviewer #1 (Recommendations for the authors):*

The whole rationale for selecting each of these tasks really needs much clearer statements and reasoning and the analysis needs to focus on the specific features of each of these tasks otherwise it is very difficult to provide a straightforward interpretation of the data.

Perhaps it is possible to provide more behavioral data to disentangle these different possibilities? In particular, providing more information on the kinetics of each task could be useful (trials to criterion for a given task, depending on prior history) if this is not limited by interindividual variability.

*Reviewer #2 (Recommendations for the authors):*

What is the rationale for assessing S1 in the optogenetic study and V1 in the calcium imaging study?

It would be helpful to have greater specificity about the areas that are being investigated. For example, are the authors looking at dygranular/granular retrosplenial cortex or both? For the more posterior site (-3.5) for retrosplenial inhibition, from the coordinates and description it appears that the area of impact would encroach into the visual cortex.

I cannot quite work out the time-line for the calcium imaging study. Is the simple experience-simple task study carried out first or is this carried out in a separate group of animals? I have concerns about the extent to which these comparisons are properly matched. There are a number of studies showing that retrosplenial in particular shows a slow development of neural representation related to spatial tasks/stimuli. That retrosplenial cortex is particularly important for familiar landmarks/cues. As such this would need to be considered with the calcium imaging. The simple experience and complex experience groups should be matched for the length of actual exposure/training with the cues allowing the possible development of spatial engrams/cell selectivity irrespective of cognitive demands. Likewise, there could be differences in terms of photobleaching depending on the timelines, number of imaging sessions. it may be that all this has been considered and taken into account but as it is currently written I can't work out these details.

The authors state there is no difference in running speed but from what I can gather they have just looked at the running speed across the entire trial. Is there any evidence of difference at the decision point – i.e., more slowing down in some conditions that could be linked to decision making?

There is a large amount of relevant literature, particularly relating to retrosplenial cortex, that the authors do not cite which could help in contextualising the current research more accurately.

For the calcium imaging the authors state they only look at the Rule A trials during the switching task – i.e., the ones that then match the simple task. I understand that provides the closest comparison across tasks but it does seem a shame not to be also looking at the Rule B data. This could provide very useful information about the specificity of the neural responses across the different rules. This could inform the extent to which these neural responses reflect specific cue/response representations or reflect more global task requirement representations and perhaps help address whether the more extensive training just helps with the encoding of the cue/responses for these specific cues or provides a better neural framework for general learning/attention.

*Reviewer #3 (Recommendations for the authors):*

These are a really interesting set of experiments that speak to an important question in the field about how prior experiences affect subsequent learning/information processing. The authors present robust data that suggest that learning the complex task (switching or delay tasks) leads to a profound effect on the role of PPC and RSC for performing a subsequent similar but different (simple) task.

One nagging question is why this is the case? I do not think more data is required for publication but it would be helpful for the authors to address the following questions either with data or further comments in the discussion/speculation portion of their manuscript.

Is this a form of schema-learning? Such that due to the prior learning of the complex task that is more dependent on PPC and RSC, the new subsequent task that is similar across many features is now being integrated with the representation of the prior complex task and thus more dependent on cortical representations? I don't think the authors need to do experiments to tease this idea apart but it would be helpful to include this in the discussion or speculation portion.

Could it be due to the timecourse of systems consolidation? I assume that more time elapses between the start of when animals learn the complex task and when they are being imaged/optogenetically manipulated on the simple task compared to the animals that just get the simple task. Meaning, animals that only get the simple task have less time experiencing these cues. If that's true, is it possible that over time, these cues/representations become more and more cortically dependent? If the authors have the data to address this question, then it would be helpful. If not, please comment on this idea in the discussion/speculation.

Is there competitive processing between the complex and simple task? Does activity in the PPC or RSC support inhibiting the prior complex task rule so that animals can better perform the simple task? If authors have the data- are there mice that perform poorly on the complex task? Does inhibition of PPC or RSC have a smaller effect on the simple task in those mice? Is there a competition between accessing/inhibiting prior tasks and the current simple task?

[Editors' note: further revisions were suggested prior to acceptance, as described below.]

Thank you for resubmitting your work entitled "Cognitive experience alters cortical involvement in goal-directed navigation" for further consideration by *eLife*. Your revised article has been evaluated by Joshua Gold (Senior Editor) and a Reviewing Editor.

The manuscript has been improved but there are some remaining issues that need to be addressed, as outlined below:

All reviewers confirmed the general interest of the paper and the high quality of the data but still felt that the paper is somewhat undermined by the lack of clarity on the conceptual framework. We suggest two lines to help address the remaining issues and improve the paper.

1. The description of the behavioral paradigm would still benefit from a revision. All reviewers felt that mentions to "navigation" and "decision-making" do not seem adequate considering the tasks that are being used do not really specifically assess these processes (navigation is at best virtual here and the task do not specifically tap into decision-making, the new take on 'decision points' – Fig1Sup1 and related text – does not really add to the story and is possibly distracting). Perhaps "visual discrimination" or "visually-cued task" or anything similar would be more relevant here (with a 'delay' and 'reversal' conditions). It is important here to edit the ms beyond simply replacing a word by another to ensure a comprehensive and consistent description of the task from abstract to discussion – this has not been done previously. When navigation was added in the revision, it was on top of decision-making, making this issue even worse. That the delay condition is not discussed in terms of working memory is still a bit puzzling (the role of the PPC could be discussed relative to that) as it operationally corresponds quite strictly to that, at least as most commonly assessed in rodent.

2. The schematics provided to explain the behavioral approach could be a little more straightforward and complete. It needs to be more explicitly stated when conditions are matched and when they are not. Based on the rebuttal from the authors, it seems that animals reportedly trained first on 'complex' tasks (that is, delay or switching) are actually trained first on the simple task so plots in Figure 1A / 2A are factually incorrect (it should be simple task  delay/switching task  simple task). This may certainly impact on the interpretation and it should be discussed: animals are thus not matched in terms of time from first exposure to stimuli and they are not matched in terms of overall exposure to stimuli and training apparatus. Perhaps considering David Smith's work could be relevant here to stress the importance of the rsc in encoding salient cues but also for late stage learning (Smith et al., Behav Neurosci, 2018). See also Vedder et al., Cereb Cortex 2017.

---

## [Author Response]

Essential revisions:1) The current narrative on training history is the unique angle to discuss the data but there are other alternative interpretations that cannot be ruled out by the present dataset. For example, one could argue that mice can form learning sets during the whole behavioral procedure and this may impact how well they perform in a given task. Alternatively, it is possible that switching from one task to another in the exact same environment is actually challenging behavioral flexibility. These are important points to consider, as all referees expressed specific concerns along these lines and you can find in the individual reviews multiple options and possibilities to address that. It is expected that if no new data is provided, substantial revisions in the introduction, discussion and possibly the title are necessary to have a more balanced analysis of the behavioral data.

We thank the reviewers for these interesting perspectives on training history effects. We want to emphasize that we do not view these suggestions as strict *alternatives* to our general conclusion that cognitive experience affects cortical task involvement. Rather, we have used ‘cognitive experience’ or ‘training history’ as umbrella terms agnostic to the underlying mechanism causing persistent cortical involvement after transition from a complex task to the simple task.

We found the reviewers’ possible explanations very interesting and have evaluated them both in the context of our experimental design and results. A key point to consider is that when mice were transitioned from a complex task to the simple task, the simple task is not a novel task. Indeed, the simple task is a well-experienced simplification of the previous task that the mouse has already encountered. Mice that are experts on the delay task have experienced the simple task, i.e. trials without a delay period, during their training procedure before being exposed to delay periods. Mice that are experts at the switching task know the simple task as one rule of the switching task and have performed according to this rule in each session prior to the task transition. Accordingly, both delay task experts and switching task experts perform at very high levels on the very first simple task session after the task transition, which we now quantify in Figure 2—figure supplement 1 (A, B). This is crucial to keep in mind when assessing ‘learning sets’ or ‘behavioral flexibility’ as possible explanations for the persistent cortical involvement after the task transitions. In classical learning sets paradigms, animals are exposed to a series of novel associations, and the learning of previous associations speeds up the learning of subsequent ones (Caglayan et al., 2021; Eichenbaum et al., 1986; Harlow, 1949). This is a distinct paradigm from ours as the simple task does not contain novel associations that are unfamiliar to the mice previously trained on the complex tasks. Relatedly, the simple task is unlikely to present a challenge of behavioral flexibility to these mice given our experimental design and the observation that mice exhibit very high performance on the first session in the simple task after the task transition. We now clarify these points in the introduction, results, and Discussion sections, also acknowledging that it will be of interest for future work to investigate how learning sets may affect cortical task involvement.

2) The calcium imaging dataset does not deliver a very clear message because the link with the functional study is not very clear (see ref 2) and no clear rationale is provided to consider only very specific portions of the data (see ref 1).

We thank the reviewers for pointing out that the relationship between the inhibition dataset and calcium imaging dataset is not clear enough. We restricted analyses of inhibition and calcium imaging data in the switching task to the identical cue-choice associations as present in the simple task (i.e. Rule A trials of the switching task). We did this because we sought to make the fairest and most convincing comparison across tasks for both datasets. However, we can now see that not reporting results with trials from the other rule causes concerns that the reported differences across tasks may only hold for a specific subset of trials.

We have now added analyses of optogenetic inhibition effects and calcium imaging results considering Rule B trials. In Figure 1—figure supplement 2, we show that when considering only Rule B trials in the switching task, effects of RSC or PPC inhibition on task performance are still increased relative to the ones observed in mice trained on and performing the simple task. We also show that overall task performance is lower in Rule B trials of the switching task than in the simple task, mirroring the differences across tasks when considering Rule A trials only.

We extended the equivalent comparisons to the calcium imaging dataset and now consider Rule B trials of the switching task in Figure 4—figure supplement 3. With Rule B trials, we still find larger mean activity and trial-type selectivity levels in RSC and PPC, but not in V1, compared to the simple task, as well as lower noise correlations. We thus find that our conclusions about area necessity and activity differences across tasks hold for Rule B trials and are not due to only considering a subset of the switching task data.

In Figure 4—figure supplement 4, we further leverage the inclusion of Rule B trials and present new analyses of different single-neuron selectivity categories across rules in the switching task, reporting a prevalence of mixed selectivity in our dataset.

Furthermore, to clarify the link between the optogenetic inhibition and the calcium imaging datasets, we have revised the motivation for the imaging dataset, as well as the presentation of its results and discussion. Investigating an area’s neural activity patterns is a crucial first step towards understanding how differential necessity of an area across tasks or experience can be explained mechanistically on a circuit level. We now elaborate on the fact that mechanistically, changes in an area’s necessity may or may not be accompanied by changes in activity within that area, as previous work in related experimental paradigms has reported differences in necessity in the absence of differences in activity (Chowdhury and DeAngelis, 2008; Liu and Pack, 2017). This phenomenon can be explained by differences in the readout of an area’s activity. We now make more explicit that in contrast to the scenario where only the readout changes, we find an intriguing correspondence between increased necessity (as seen in the inhibition experiments) and increased activity and selectivity levels (as seen in the imaging experiments) in cortical association areas depending on the current task and previous experience. Rather than attributing the increase in necessity solely to these observed changes in activity, we highlight that in the simple task condition already, cortical areas contain a high amount of task information, ruling out the idea that insufficient local information would cause the small performance deficits from inhibition. Our results thus suggest that differential necessity across tasks and experience may still require changes at the readout level despite changes in local activity. We view our imaging results as an exciting first step towards a mechanistic understanding of how cognitive experience affects cortical necessity, but we stress that future work will need to test directly the relationship between cortical necessity and various specific features of the neural code.

3) On the theoretical ground, authors need to better explain why these 3 particular tasks were selected and why they put such a strong emphasis on "decision-making" (see ref 1 and 2). These tasks do not seem to correspond to classic "decision-making" tasks. The take on decision-making is thus unclear and needs to be justified or removed to provide a fair and accurate description of each task.

We thank the reviewers for pointing this out and acknowledge that the previous emphasis on decisionmaking may have created expectations that we demonstrate effects that are specific to the ‘decisionmaking’ aspect of a decision task in contrast to sensory processing or motor output. As we do not isolate the decision-making process specifically, we have substantially revised our wording around the tasks and removed the emphasis on decision-making, including in the title. Rather than decision-making, we now highlight the navigational aspect of the tasks employed.

Furthermore, we now have elaborated on the motivation for the specific tasks in this study, on their cognitive requirements, and on behavioral quantifications indicating how mice solve the different tasks. We used the paradigm of spatial navigation as an ethologically relevant behavior in which mice use sensory cues to navigate to hidden reward locations associated with those cues. We chose PPC as one of the main areas studied here because there is a long history of literature on PPC in the context of decisionmaking across species, generally proposing that PPC is thought to convert sensory cues into actions (Freedman and Ibos, 2018; Goard et al., 2016; Harvey et al., 2012). Moreover, several studies have highlighted the importance of PPC for navigation-based decision tasks (Driscoll et al., 2017; Harvey et al., 2012; Pinto et al., 2019). Importantly, a recent comparison of several inactivation studies in rodents proposed that specific task components involving short-term memory may cause PPC to be necessary for a given decision task (Lyamzin and Benucci, 2019). In choosing what complex tasks to use for establishing more complex cognitive experience in comparison to the simple task, these task components represented concrete modifications of the simple task to use, one of which was the insertion of a delay period. Indeed, a delay period between the sensory cue and ultimate choice point or choice report is a common design in decision tasks, and this task has previously been shown to require PPC activity (Driscoll et al., 2017; Harvey et al., 2012; Pinto et al., 2019). It thus seemed like an obvious ‘complex task’ candidate to compare the simple task to.

We also employed another complex task termed the switching task in this study for several reasons. First, the switching task allowed us to compare the same, visually identical trial types across tasks, offering a clean experimental comparison. Second, we wanted to understand whether a short-term memory component across trials as is necessary in the switching task, as opposed to a within-trial component like in the delay task, would also cause increased cortical involvement, thereby testing and expanding on the task parameters hypothesized to underlie PPC involvement (Lyamzin and Benucci, 2019). Third, this additional task enabled us to probe whether the effect of cognitive experience on cortical involvement was a more general phenomenon, or one that was rather specific to experience of delay periods.

Finally, while we removed the emphasis on the decision-making process in our tasks, the reviewers’ comments made us realize that it seems illuminating to infer ‘decision points’ in the tasks employed here, that is how soon a mouse’s ultimate choice can be decoded from its running pattern as it progresses through the maze towards the Y-intersection. We now show these results in Figure 1—figure supplement 1. Interestingly, we found that in the delay task, choice decoding accuracy was already very high during the cue period before the onset of the delay. Nevertheless, we had shown that overall task performance and performance with inhibition were lower in the delay task compared to the simple task. Also, in segment-specific inhibition experiments, we had found that inhibition during only the delay period still decreased task performance substantially more than in the simple task. Thus, how early a mouse made its ultimate decision does not appear predictive of the inhibition-induced task decrements, which we also directly quantify in Figure 1—figure supplement 1.

Reviewer #1 (Recommendations for the authors):The whole rationale for selecting each of these tasks really needs much clearer statements and reasoning and the analysis needs to focus on the specific features of each of these tasks otherwise it is very difficult to provide a straightforward interpretation of the data.

We have followed this recommendation in detail. Please see our reply to point 1 above.

Perhaps it is possible to provide more behavioral data to disentangle these different possibilities? In particular, providing more information on the kinetics of each task could be useful (trials to criterion for a given task, depending on prior history) if this is not limited by interindividual variability.

In addition to our detailed reply to point 2 above, we thank the reviewer for this specific suggestion. In our task training procedures, all trials were initially ‘visually guided trials’, where the correct choice was signaled with a checkerboard at the end of the correct Y-arm. This was especially crucial during initial learning of the switching task. To keep training procedures as similar as possible across tasks, mice trained on the simple task also learned the cue-choice associations initially with visually guided trials. Throughout training, the fraction of “visually guided trials” was gradually reduced on a session-by-session basis by the experimenter, based on the mouse’s previous performance. Unfortunately, this training procedure precludes a fair comparison between de novo learning of the simple task and simple task performance in mice transitioned to the simple task from a complex task. Instead, we quantify the performance of the mouse on the very first session in the simple task after a transition from a complex task. We find that mice perform at very high levels even on the first session in the simple task after this transition, which we now report in Figure 2—figure supplement 1. This new analysis allows us to assess (and rule out) the hypothesis that the transition to the simple task presents a challenge of cognitive flexibility, as explained in detail above.

Reviewer #2 (Recommendations for the authors):What is the rationale for assessing S1 in the optogenetic study and V1 in the calcium imaging study?

We thank the reviewer for pointing out a lack of clarity on this in the previous version. In the optogenetic experiments, we chose S1, an area not implicated in visual decision-making, as a control spot size-matched to the PPC inhibition spot to see if the effects of task complexity and cognitive experience would be completely generalized across cortical areas. The small effects of S1 inhibition on task performance revealed that this was not the case. We did not choose V1 as a target because inhibition of V1 would likely cause visual processing deficits in all visual navigation tasks regardless of cognitive experience. From unpublished data from our lab in a different but related behavioral task, we find that inhibiting V1 using the optogenetics methods used in this paper results in behavior consistent with the mouse not being able to see the visual virtual environment. Mice end up spinning in the virtual environment and are unable to complete trials. We did not feel that V1 inhibition would be very informative for that reason.

In the calcium imaging experiments, we chose to include V1 because it is densely interconnected with PPC and RSC (Zhang et al., 2016), contains cognitive, non-visual signals in navigation-based decision tasks (Koay et al., 2020; Saleem et al., 2018), and is likely a key contributor during visual navigation. We acknowledge that imaging in S1 as well would have given additional insight into how localized the observed differences in neural selectivity across tasks and experiences are to RSC and PPC, but a lack of difference in V1 already established that this is not a general phenomenon across all cortical areas. We have clarified our motivation for S1 and V1 inclusion in different experiments in the Results section.

It would be helpful to have greater specificity about the areas that are being investigated. For example, are the authors looking at dygranular/granular retrosplenial cortex or both? For the more posterior site (-3.5) for retrosplenial inhibition, from the coordinates and description it appears that the area of impact would encroach into the visual cortex.

We apologize that we did not explicitly specify which parts of RSC were targeted across experiments beyond providing the coordinates. In the inhibition experiments, we targeted the entire RSC including both dysgranular and granular areas and spanning the entire anterior-posterior extent. In the imaging experiments, as we imaged other cortical areas simultaneously with RSC, we were limited to imaging a subregion of RSC to maintain sufficient temporal resolution. We targeted the anterior part of dysgranular RSC. V1 imaging was performed in the most anterior part of V1. We have now provided this information in the Methods section.

As to possible off-target effects during the inhibition experiments, we cannot exclude that a small part of medial V1 may have been affected. However, previous work using similar experimental conditions has reported that inhibition of posterior RSC resulted in different behavioral effects than inhibition of medial V1 (Pinto et al., 2019), making it unlikely that the effects of RSC inhibition we report here can be accounted for by affecting V1.

I cannot quite work out the time-line for the calcium imaging study. Is the simple experience-simple task study carried out first or is this carried out in a separate group of animals? I have concerns about the extent to which these comparisons are properly matched. There are a number of studies showing that retrosplenial in particular shows a slow development of neural representation related to spatial tasks/stimuli. That retrosplenial cortex is particularly important for familiar landmarks/cues. As such this would need to be considered with the calcium imaging. The simple experience and complex experience groups should be matched for the length of actual exposure/training with the cues allowing the possible development of spatial engrams/cell selectivity irrespective of cognitive demands. Likewise, there could be differences in terms of photobleaching depending on the timelines, number of imaging sessions. it may be that all this has been considered and taken into account but as it is currently written I can't work out these details.

We thank the reviewer for pointing out missing information on task learning and when the imaging sessions took place in relation to that. We have now added quantifications of both to Figure 4—figure supplement 1. We show that in the imaging dataset, like in the inhibition dataset, learning of the simple task is much faster than learning of the switching task (panel B), which is performed by different groups of mice. Indeed, mice with switching task experience that were transitioned to the simple task have overall been trained for a longer period of time than mice with simple task experience only. In panel C, we now directly quantify the number of sessions that the mice had experienced until a given imaging session, illustrating that we did not match the number of sessions across experiences for the different groups of mice performing the simple task. However, we have various reasons to believe that it is unlikely for this difference to cause the observed differences across experiences in cortical necessity or activity patterns. First, inhibition effects are stable across long periods of times, even over weeks, which we now demonstrate in Figure 1—figure supplement 1. Second, mice in the ‘simple task with only simple task experience’ condition had extensive experience in the visual cues and the task by the time they were imaged. They had experienced the visual cues from the very beginning of VR training, including the first linear maze training stage which typically lasted 7-14 sessions. We had failed to specify the duration of this training stage but have now added it to the Methods section. Furthermore, while the mice reached high performance levels after less than 10 sessions of simple task training (Figure 4—figure supplement 1, panel B), we now show that the imaging sessions in these mice occurred on average only after around 20 sessions of overall simple task experience (panel C). Thus, mice had experienced the simple task for about three weeks by the time of imaging, and the visual cues themselves for another 1-2 weeks before. This amount of cue and task experience is well in line with previous work on the representation of spatial landmarks in RSC. For example (Fischer et al., 2020) report 2-4 week training periods preceding imaging in expert mice, and (Milczarek et al., 2018) reported the emergence of stable context-specific engrams in RSC over a period of 3 weeks.

As to the possibility of photobleaching, we acknowledge that this may be an issue if especially high laser powers need to be employed, which was not the case in our study. Further, we would expect that if photobleaching affected neural activity levels, it would lead to lower measurements of activity and trialtype selectivity with long-term laser exposure, thus decreasing such measures in mice with more imaging sessions, i.e. mice imaged in the simple task that we previously had imaged in the switching task. However, in those mice we reported higher activity and trial-type selectivity levels compared to mice imaged in the simple task without switching task experience, who had experienced fewer imaging sessions. It thus seems highly unlikely that the effects we observe here stem from photobleaching.

The authors state there is no difference in running speed but from what I can gather they have just looked at the running speed across the entire trial. Is there any evidence of difference at the decision point – i.e., more slowing down in some conditions that could be linked to decision making?

We thank the reviewer for this interesting suggestion of characterizing decision points. We have now included additional analyses in Figure 1—figure supplement 1 where we ask how soon we can decode a mouse’s ultimate choice from its running pattern as it progresses down the maze. Across tasks, we found that choice decoding accuracy rose early in the maze, especially early on in the delay task, even during the cue period before the onset of the delay. Nevertheless, we had shown that overall task performance and performance with inhibition were lower in the delay task compared to the simple task. Also, in segment-specific inhibition experiments (now in Figure 1—figure supplement 4), we had found that inhibition during only the delay period or only the cue period decreased task performance substantially more than in the simple task, thus finding an interesting absence of differential inhibition effects around decision points in the delay task.

Based on the reviewers’ shared issues around our previous emphasis on decision-making, we decided to remove the focus on decision-making instead of pursuing further analyses in this direction. We have substantially revised our wording around the tasks and now highlight the navigational aspect of the tasks employed, including in the title.

There is a large amount of relevant literature, particularly relating to retrosplenial cortex, that the authors do not cite which could help in contextualising the current research more accurately.

We apologize that we had not contextualized our findings appropriately, particularly for RSC function. We have now extended the introduction section to better motivate the particular areas studied here, including additional citations on RSC. We introduce that RSC is poised to enable the transformation between egocentric and allocentric reference frames and to support spatial memory across various timescales (Alexander et al., 2020; Alexander and Nitz, 2015; Fischer et al., 2020; Milczarek et al., 2018; Pothuizen et al., 2009; Powell et al., 2017). It furthermore has been shown to be involved in cognitive processes beyond spatial navigation, such as such as context-dependent cue-choice associations, temporal learning and value coding (Franco and Goard, 2021; Hattori et al., 2019; Todd et al., 2015), making it an especially interesting area to study in the context of cognitive experience. We have furthermore expanded the Discussion section to explain the implications of our findings not just for cortical function in general, but also for PPC and RSC individually. We now discuss that increased RSC necessity in more complex tasks as well as due to complex task experience further support the emerging view that RSC plays a crucial role in cognitive aspects of navigation decisions, integrating visual cues flexibly with memory signals to plan navigational routes or choices (Franco and Goard, 2021; Spiers and Maguire, 2006; Stacho and Manahan-Vaughan, 2022).

If the reviewer has in mind other specific references or findings that are important to cite or mention regarding RSC, we would be happy to include them.

For the calcium imaging the authors state they only look at the Rule A trials during the switching task – i.e., the ones that then match the simple task. I understand that provides the closest comparison across tasks but it does seem a shame not to be also looking at the Rule B data. This could provide very useful information about the specificity of the neural responses across the different rules. This could inform the extent to which these neural responses reflect specific cue/response representations or reflect more global task requirement representations and perhaps help address whether the more extensive training just helps with the encoding of the cue/responses for these specific cues or provides a better neural framework for general learning/attention.

We acknowledge that not reporting results with trials from the other rule caused concerns that the reported differences across tasks may only hold for a specific subset of trials. We have now added analyses of both optogenetic inhibition effects and calcium imaging results considering Rule B trials. In Figure 1— figure supplement 2, we show that when considering only Rule B trials in the switching task, effects of RSC or PPC inhibition on task performance are still increased relative to the ones observed in mice trained on and performing the simple task. We also show that overall task performance is lower in Rule B trials of the switching task than in the simple task, mirroring the differences across tasks when considering Rule A trials only.

We extended the equivalent comparisons to the calcium imaging dataset, only considering Rule B trials of the switching task in Figure 4—figure supplement 3. With Rule B trials only, we still find larger mean activity and trial-type selectivity levels in RSC and PPC, but not in V1, compared to the simple task, as well as lower noise correlations. We thus find that our conclusions about area necessity and activity differences across tasks hold for Rule B trials and are not due to only considering a subset of the switching task data.

Finally, following the reviewer suggestion above, in Figure 4—figure supplement 4, we further leverage the inclusion of Rule B trials and present new analyses of different single-neuron selectivity categories across rules in the switching task. Interestingly, we find low rule selectivity across the population (panel A), but a prevalence of mixed selectivity in our dataset (panels B, C) (Rigotti et al., 2013), that is neurons with selectivity to a single cue-choice combination instead of general selectivity to the cue, choice, or rule. Overall, we found a high degree of specificity in neuronal responses rather than general modulation of activity in one rule over the other. More detailed analyses of activity in the switching task seem beyond the scope of this paper, and other ongoing work in the lab is investigating behavioral strategies and neural coding properties in the switching task in more detail.

Reviewer #3 (Recommendations for the authors):These are a really interesting set of experiments that speak to an important question in the field about how prior experiences affect subsequent learning/information processing. The authors present robust data that suggest that learning the complex task (switching or delay tasks) leads to a profound effect on the role of PPC and RSC for performing a subsequent similar but different (simple) task.

We are glad to see that the reviewer finds our main finding interesting and convincing.

One nagging question is why this is the case? I do not think more data is required for publication but it would be helpful for the authors to address the following questions either with data or further comments in the discussion/speculation portion of their manuscript.

We thank the reviewer for raising these points. We have addressed each of the specific comments below with new analyses and discussion text. We have also added a paragraph to the Discussion section of the paper that is directly related to this topic.

Is this a form of schema-learning? Such that due to the prior learning of the complex task that is more dependent on PPC and RSC, the new subsequent task that is similar across many features is now being integrated with the representation of the prior complex task and thus more dependent on cortical representations? I don't think the authors need to do experiments to tease this idea apart but it would be helpful to include this in the discussion or speculation portion.

We thank the reviewer for this suggestion and have carefully evaluated this hypothesis. The main point to consider is that for mice transitioned from either of the complex tasks to the simple task, the simple task is not a novel task, but rather a well-known simplification of the previous tasks. Mice that are experts on the delay task have experienced the simple task, i.e. trials without a delay period, during their training procedure before being exposed to delay periods. Switching task expert mice know the simple task as one rule of the switching task and have performed according to this rule in each session prior to the task transition. Accordingly, upon the transition to the simple task, both delay task expert mice and switching task expert mice perform at very high levels on the very first simple task session. We now quantify and report this in Figure 2—figure supplement 1 (A, B).

This is crucial to keep in mind when assessing schema learning as a possible explanation for the persistent cortical involvement after the task transitions. In many classical schema learning or related ‘learning sets’ paradigms, animals are exposed to a series of novel associations, and the learning of previous associations has created a mental schema into which subsequent novel associations are readily incorporated (Bartlett, 1932; Caglayan et al., 2021; Eichenbaum et al., 1986; Harlow, 1949; McKenzie et al., 2013; Piaget, 1926; Tse et al., 2007). This is a distinct paradigm from ours because the simple task does not contain novel associations that are new to the mice already trained on the complex tasks. Relatedly, the simple task is unlikely to present a challenge of behavioral flexibility to these mice given our experimental design and the observation of high simple task performance in the first session after the task transition. We now introduce and discuss these points in the introduction, results and Discussion sections.

Could it be due to the timecourse of systems consolidation? I assume that more time elapses between the start of when animals learn the complex task and when they are being imaged/optogenetically manipulated on the simple task compared to the animals that just get the simple task. Meaning, animals that only get the simple task have less time experiencing these cues. If that's true, is it possible that over time, these cues/representations become more and more cortically dependent? If the authors have the data to address this question, then it would be helpful. If not, please comment on this idea in the discussion/speculation.

We thank the reviewer for this hypothesis, which we have evaluated on theoretical ground and with additional analyses. In Figure 1—figure supplement 1, we have now correlated the number of experienced sessions in each task with the task performance decrements from cortical inhibition. We found that inhibition effects were stable across long periods of times, even over weeks, in all tasks including the simple task, which we now mention in the Discussion section. Our inhibition dataset thus does not support the idea of incomplete systems consolidation accounting for small inhibition effects on performance in the simple task without preceding complex task experience.

In presentation of the calcium imaging dataset, we apologize for previously missing information on task learning and when the imaging sessions took place in relation to that. We have now added quantifications of both to Figure 4—figure supplement 1. We show that in the imaging dataset, like in the inhibition dataset, learning of the simple task is much faster than learning of the switching task (panel B), which is performed by different groups of mice. Indeed, mice with switching task experience that were transitioned to the simple task have overall been trained for a longer period of time than mice with simple task experience only. In panel C, we now quantify the number of sessions that the mice had experienced until a given imaging session, illustrating that we did not match the number of sessions across experiences for the different groups of mice performing the simple task. However, mice in the ‘simple task with only simple task experience’ condition had extensive experience in the visual cues and the task by the time they were imaged. They had experienced the visual cues from the very beginning of VR training, including the first linear maze training stage which typically lasted 7-14 sessions. We had failed to specify the duration of this training stage but have now added it to the Methods section. Furthermore, while the mice reached high performance levels after less than 10 sessions of simple task training (Figure 4—figure supplement 1, panel B), we now show that the imaging sessions in these mice occurred on average only after around 20 sessions of overall simple task experience (panel C). Thus, mice had experienced the simple task for about three weeks by the time of imaging, and the visual cues themselves for another 1— 2 weeks before. This amount of cue and task experience is well in line with previous work on the representation of spatial landmarks in RSC. For example (Fischer et al., 2020) report 2-4 week training periods preceding imaging in expert mice, and (Milczarek et al., 2018) reported the emergence of stable context-specific engrams in RSC over a period of 3 weeks. Therefore, it overall seems unlikely that incomplete systems consolidation would account for lower mean activity and trial-type selectivity in mice performing the simple task without complex task experience.

Is there competitive processing between the complex and simple task? Does activity in the PPC or RSC support inhibiting the prior complex task rule so that animals can better perform the simple task? If authors have the data- are there mice that perform poorly on the complex task? Does inhibition of PPC or RSC have a smaller effect on the simple task in those mice? Is there a competition between accessing/inhibiting prior tasks and the current simple task?

We thank the reviewer for raising this interesting possibility. The overall high control performance levels in both the delay task and the switching task across mice may make it challenging or impossible to see a possible relationship between each mouse’s prior complex task performance and subsequent performance decrements from RSC or PPC inhibition in the simple task after the task transition. We nevertheless followed the reviewer’s suggestion and looked for such a possible signature of competition. We did not see a systematic relationship between these two variables across areas and tasks, though perhaps an interesting trend in PPC in the switching task-to-simple task transition. We imagine that future studies with larger sample sizes and larger ranges of performance in the complex task would be better suited to address this question, so we chose not to include plots in Author response image 1 in the revised manuscript.

**Author response image 1. sa2fig1:** 

Further behavioral data that seem relevant to the question of competition are presented in what is now Figure 2—figure supplement 1. There we show that mice initially trained on the switching task could still perform the switching task even after two weeks of only simple task experience, i.e. they reached high performance levels on the long unseen rule within a single session after re-exposure to the switching task (panels D-E). This suggests that if the switching task was indeed inhibited during performance of the simple task, this inhibition or competition could not have been permanent.[Editors' note: further revisions were suggested prior to acceptance, as described below.]

All reviewers confirmed the general interest of the paper and the high quality of the data but still felt that the paper is somewhat undermined by the lack of clarity on the conceptual framework. We suggest two lines to help address the remaining issues and improve the paper.1. The description of the behavioral paradigm would still benefit from a revision. All reviewers felt that mentions to "navigation" and "decision-making" do not seem adequate considering the tasks that are being used do not really specifically assess these processes (navigation is at best virtual here and the task do not specifically tap into decision-making, the new take on 'decision points' – Fig1Sup1 and related text – does not really add to the story and is possibly distracting). Perhaps "visual discrimination" or "visually-cued task" or anything similar would be more relevant here (with a 'delay' and 'reversal' conditions). It is important here to edit the ms beyond simply replacing a word by another to ensure a comprehensive and consistent description of the task from abstract to discussion – this has not been done previously. When navigation was added in the revision, it was on top of decision-making, making this issue even worse. That the delay condition is not discussed in terms of working memory is still a bit puzzling (the role of the PPC could be discussed relative to that) as it operationally corresponds quite strictly to that, at least as most commonly assessed in rodent.

We thank the reviewers for providing this feedback and identifying a point that needs clarification. We continue to think that “navigation” and “decision-making” are appropriate descriptors of our behavioral paradigm. We realize now that it will be helpful to explicitly state our reasoning on this point in the manuscript. Our task is a two-alternative forced choice task, which is a common paradigm used to study decision-making. In our task, the mouse must discriminate between two visual cues and then make a choice to run to the right or the left in a virtual reality Y-maze. This design is similar in concept to some of the classic decision-making tasks, such as the random dot motion task. In the random dot motion task, a monkey must discriminate between the motion of the stimulus in one direction versus another and then report a choice with a saccadic eye movement to the left or the right. While it is true that the visual discrimination in our task is a simple one between two very different stimuli, the task nevertheless captures the main components of making a decision and has similar design to classic decision-making tasks. That is, our task requires mice to choose between two options based on a visual cue. We therefore feel that “decision-making” is an appropriate descriptor. We also feel that “navigation” is appropriate as the paradigm of navigation in virtual environments has been used to study many of the key elements of navigation circuits, including place cells, grid cells, and head direction cells, across a wide range of species, including mice, rats, flies, humans, and monkeys. Multiple papers using VR have reproduced key aspects of navigation cells and circuits found in freely moving animals and have yielded important new insights into the mechanisms of spatial navigation. Given the long list of papers that have used virtual reality to study navigation, we feel that “navigation” is an appropriate term to use. We note that we and others have used the terms “navigation” and “decision-making” in multiple previous papers to describe similar behavioral tasks. There is thus a strong precedent for using these terms for these tasks in the literature. We remain unsure why the reviewers consider these terms not appropriate. However, we feel they are appropriate descriptors that follow the terminology used in the literature, and thus we do not feel it is necessary to remove or change these terms.

We agree with the reviewers that it might be helpful to define what we mean by “navigation” and “decision-making” in relation to our behavioral paradigm. We have therefore added sentences to the introduction section that provide this clarification. While this topic could be discussed in great length in the paper, to keep the introduction as succinct as possible, we limited to the new sentences to only brief definitions. In the introduction, we now write: “Specifically, we used a virtual reality Y-maze in which mice encounter one of two visual cues and use the cue to virtually navigate to the left or right Y-arm to receive a reward (Harvey et al., 2012). This paradigm involves decision-making because it requires mice to choose one of two Y-arms based on a learned association with a visual cue. Furthermore, it involves navigation because mice must execute a sequence of movements to reach a goal location in a virtual environment, and movement through virtual mazes is known to recruit navigation-related circuits (Aronov and Tank, 2014; Domnisoru et al., 2013; Harvey et al., 2009). We therefore refer to our paradigm as goal-directed navigation, which implicitly includes decision-making to choose a goal location.” (lines 57-64).

Regarding the delay task, we agree with the reviewers that terminologies for working memory and delay tasks are often used interchangeably and that delay tasks are typically thought to require working memory. However, as mentioned in our previous response, we have concerns about using the term “working memory” in our specific case because it is not clear how the mice actually solve the task. We report that, in the delay task, mice start to turn to the right or left before the delay segment starts. It is possible that the mice might simply continue running to the right or the left through the delay without actually remembering the identity of the cue. While we think this possibility is unlikely, we cannot formally rule it out with this paper’s dataset. Rather than using a term that suggests some interpretation of how the mice solve the task, we instead prefer to use a naming of the task that is descriptive of the task design.

2. The schematics provided to explain the behavioral approach could be a little more straightforward and complete. It needs to be more explicitly stated when conditions are matched and when they are not. Based on the rebuttal from the authors, it seems that animals reportedly trained first on 'complex' tasks (that is, delay or switching) are actually trained first on the simple task so plots in Figure 1A / 2A are factually incorrect (it should be simple task  delay/switching task  simple task). This may certainly impact on the interpretation and it should be discussed: animals are thus not matched in terms of time from first exposure to stimuli and they are not matched in terms of overall exposure to stimuli and training apparatus. Perhaps considering David Smith's work could be relevant here to stress the importance of the rsc in encoding salient cues but also for late stage learning (Smith et al., Behav Neurosci, 2018). See also Vedder et al., Cereb Cortex 2017.

We thank the reviewers for pointing out this issue that needs clarification. The schematics in Figures 1A and 2A are factually correct, but they are indeed incomplete descriptions of the training procedure for each set of mice. We prefer to use these schematics because they highlight the main difference between the two groups of mice being compared. It would be too complex to show the complete training procedure in the schematics in the main figures. We have therefore added a new supplementary figure (Figure 1—figure supplement 1) that shows the complete training procedure for each set of mice. We note that it would be incomplete to show “simple task -> delay/switching task -> simple task” as suggested by the reviewers because the mice experienced a training step even before the simple task. Also, it is factually incorrect to show “simple task -> switching task -> simple task” because this training sequence was not used. The mice in the switching task had experience in the simple task in the sense that one rule of the switching task is identical to the simple task. However, these mice were never trained on the simple task alone before experiencing the switching task. In addition, in this new supplementary figure, we have included the average number of training days spent at each step. We hope this will clearly illustrate to readers that the mice studied on different tasks had undergone different numbers of training sessions on average. In addition to this new supplementary figure, we note that in the revision, we added multiple sentences to the paper that specifically talk about the training procedure and the differences in training times (lines 115-117, 210-212, 485-495). We hope these lines, the new supplementary figure, and the description of mouse training in the methods section collectively provide a transparent description of the experience of each mouse.

Thank you for the suggestion to mention the work from David Smith and others. We have added citations to the suggested papers in the Discussion section, as further evidence that our task training and exposure times preceding calcium imaging or optogenetic perturbations were well in line with or even exceeded previous work.